# Unravelling earth flow dynamics with 3D time series derived from UAV-SfM models

François Clapuyt[1], Veerle Vanacker[1], Fritz Schlunegger[2], Kristof Van Oost[1]

[1]Earth and Life Institute, Georges Lemaître Centre for Earth and Climate Research, Université Catholique de Louvain, Belgium.

[2]Institut für Geologie, Universität Bern, Switzerland

*Correspondence to:* François Clapuyt (francois.clapuyt@uclouvain.be)

**Abstract.** Accurately assessing geohazards and quantifying landslide risks in mountainous environments gain importance in the context of the on-going global warming. For an in-depth understanding of slope failure mechanisms, accurate monitoring of the mass movement topography at high spatial and temporal resolutions remains essential. The choice of the acquisition framework for high-resolution topographic reconstructions will mainly result from the trade-off between the spatial resolution needed and the extent of the study area. Recent advances in the development of UAV-based (Unmanned Aerial Vehicle) image acquisition combined with Structure-from-Motion (SfM) algorithm for 3-dimensional (3D) reconstruction makes the UAV-SfM framework a competitive alternative to other high-resolution topographic techniques.

In this study, we aim at getting an in-depth knowledge of the Schimbrig earthflow located in the foothills of the Central Swiss Alps, by monitoring ground surface displacements at very high spatial and temporal resolution using the efficiency of the UAV-SfM framework. We produced distinct topographic datasets for three acquisition dates between 2013 and 2015 in order to conduct a comprehensive 3D analysis of the landslide. Therefore, we computed (1) the sediment budget of the hillslope, and (2) the horizontal and (3) the 3-dimensional surface displacements, and. The multitemporal UAV-SfM based topographic reconstructions allowed us to quantify rates of sediment redistribution and surface movements. Our data show that the Schimbrig earthflow is very active with mean annual horizontal displacement ranging between 6 and 9 meters. Combination and careful interpretation of high-resolution topographic analyses reveal the internal mechanisms of the earthflow and its complex rotational structure. In addition to variation in horizontal surface movements through time, we interestingly showed that the configuration of nested rotational units changes through time. Although there are major changes in the internal structure of the earthflow in the 2013-2015 period, the sediment budget of the drainage basin is nearly in equilibrium. As a consequence, our data show that the time lag between sediment mobilization by landslides and enhanced sediment fluxes in the river network can be considerable.

**1 Introduction**

The northern Alpine border, and particularly the foothills of the Central European Alps of Switzerland are prone to slope instabilities, with 6% of its surface area affected by landslide hazards (Lateltin et al., 2005). The outcrop of landslide-prone formations of Flysch and Molasse bedrock have thick interbedded mudstones, which increase conditions for landslide occurrence. This led local authorities to draw particular attention to georisk management by identifying and assessing landslide hazards (Raetzo et al., 2002) to minimize socio-economic impacts through loss of human life and damage to infrastructure. Climate change is likely to increase landslide hazards in a near future (Crozier, 2010; Huggel et al., 2012). The on-going global warming is characterized by higher mean, minimum and maximum air temperatures and more frequent precipitation events (Cubasch et al., 2013), which has in turn an influence on pre-conditions and triggering mechanisms for landslides (Bennett et al., 2013; Crozier, 2010). The increasing trend in intense precipitation reported for the Northern part of Switzerland is likely to change soil moisture conditions at the Alpine foothills during autumn and spring (Seneviratne et al., 2010).

The recent advances in high-resolution topography representation techniques and acquisition platforms, combined with decreasing surveying costs, has led to an increasing availability of topographic datasets over the last decade. High-resolution topographic reconstructions of the earth's landforms are nowadays used for accurate topographic representation and modelling in soil science, volcanology, glaciology, river and coastal morphology (Tarolli, 2014). The study of geo-hazards benefits from these technological advances for proper monitoring of surface displacements and topographic deformation (Caduff et al., 2015; Jaboyedoff et al., 2012; Joyce et al., 2009; Metternicht et al., 2005; Scaioni et al., 2014). Measurements of surface deformation can be retrieved from chronological sequences of sub-meter resolution topography. The choice of the acquisition framework for topographic representations will always result from the trade-off between (1) the spatial resolution needed and (2) the extent of the study area (Passalacqua et al., 2015). In the specific case of repeated measurements of mass movement and hillslope processes in mountainous environments, the following three parameters are narrowing down the possibilities for the acquisition framework: (3) the surveying cost, (4) the necessary return period for proper monitoring, and (5) the accessibility of the study area.

In the past, the evolution of landslides was studied based on aerial photographs, which were either used for visual interpretation or photogrammetric reconstruction of digital elevation models (DEM), characterized by an up-to several-meter accuracy over regional spatial extents (e.g. Casson et al., 2003; Guns and Vanacker, 2014; Hervás et al., 2003; Prokešová et al., 2010; Vanacker et al., 2003; van Westen and Lulie Getahun, 2003). Historical aerial photographs allow to reconstruct surface displacements over the last fifty years when data availability is optimal (Prokešová et al., 2010; van Westen and Lulie Getahun, 2003). However, the use of aerial photographs for monitoring dynamic environments is often limited by the relatively large time interval between flight campaigns along with accuracy and resolution, that does not match current standards for topographic representations. Optical satellite images, i.e. ASTER, Quickbird or Pléiades, have successfully been used for

landslide monitoring with decimeter accuracy by applying optical image correlation techniques on time series of satellite images (e.g. Delacourt et al., 2004; Kääb, 2002; Stumpf et al., 2017). Even though sub-meter resolution images such as Pléiades are now used, the price of very-high resolution optical satellite imagery might remain an obstacle for long-term monitoring of dynamic environments that require frequent monitoring and dense time series. Surface deformation is now increasingly monitored using radar technology, that is Synthetic Aperture Radar, i.e. SAR, and Light Detection and Ranging, i.e. LIDAR. These techniques can be deployed on airborne and ground-based platforms that have distinct inherent accuracies and ranges (Passalacqua et al., 2015). Terrestrial or satellite SAR and airborne LIDAR (ALS) can cover areas ranging in size from 1 to thousands of kilometres for spacecraft-embedded SAR, at a metric spatial resolution. Ground-based LIDAR, i.e. terrestrial laser scanning (TLS), can provide topographic data of areas ranging in size from 0.01 to a few square-kilometres at centimetre spatial resolution. For complex topography, multiple laser scans from different viewpoints are often required to cover the surface with reasonable overlap. In hummocky topography, the viewpoint's visibility fields can be particularly narrow. Spacecraft-embedded SAR data, e.g. TerraSAR-X and ALOS/PALSAR images, enable to track displacements of mass movements characterized by important variations in vegetation cover and soil surface between the acquisition dates (e.g. Raucoules et al., 2013; Schlögel et al., 2015). Aerial and ground-based LIDAR datasets have mainly been used for landslide monitoring using DEM differencing (e.g. Blasone et al., 2014; Ventura et al., 2011; Wheaton et al., 2010) and image correlation (Travelletti et al., 2014). The spatial or temporal resolution and/or high acquisition cost have so far limited the applicability of traditional photogrammetric reconstructions, high-resolution optical satellite imagery and aerial- and satellite-based laser scanning for multi-temporal 3D analyses of landslide-prone areas.

A recent technique to reconstruct very-high resolution topography is the image-based Structure from Motion (SfM) algorithm. Based on pictures taken from a standard camera, 3-dimensional (3D) topographic reconstructions can cover up to several km-wide study areas if the camera is carried by an Unmanned Aerial Vehicle, i.e. UAV (Immerzeel et al., 2014; Lucieer et al., 2014; Rippin et al., 2015). UAV-SfM has similar performance as TLS data, i.e. decimetre to centimetre accuracy output (Clapuyt et al., 2016). Compared to other topographic acquisition techniques, the UAV-SfM framework is low-cost and flexible in its implementation, and particularly attractive for survey campaigns in poorly accessible landslide-prone terrain. The UAV-SfM technique is now widely applied for single 3D topographic reconstructions and mapping of natural environments (Eltner et al., 2016). However, the potential of the SfM algorithm and the derived 3D dense point clouds is not yet fully exploited for landslide monitoring. Time-series of topographic data derived from UAV-SfM data are mainly used to compute DEMs of differences (DoD) from temporal pairs of topographic representations (Fernández et al., 2016; Huang et al., 2017; Peternel et al., 2017; Tanteri et al., 2017). DEM differencing has been combined with image correlation techniques (COSI-Corr) to assess horizontal displacements and determine landslide dynamics (Lucieer et al., 2014; Turner et al., 2015). Recent UAV-based landslide studies included the assessment of point-based horizontal displacements from displacement vectors between corresponding features on pairs of orthophotos (Fernández et al., 2016;Peternel et al., 2017), and cross-section profiles to get an insight of the landslide's internal structure (Tanteri et al., 2017). Using ground-based images, Stumpf et al.

(2014) assessed the seasonal dynamics of the Super-Sauze landslide over a 2-year period, by using a 3D change detection algorithm, i.e. Multiscale Model to Model Cloud Comparison (M3C2).

This paper aims to take benefit from time series of UAV-SfM derived topographic representations at very-high spatial
resolution and evaluate the application of low-cost and lightweight methods to infer internal landslide dynamics in mountainous environments. As such, UAV-SfM models can complement data on the internal structure and mass movements' dynamics that are commonly derived from geophysical and/or geotechnical methods (e.g. Fressard et al., 2016; Merritt et al., 2014; Perrone et al., 2014).

This study differs from previous work that applied the UAV-SfM framework on landslide prone areas. First, the accuracy and reproducibility of the UAV-SfM based topographic representations is directly accounted for in the change detection. Second, the landslide kinematics and dynamics are reconstructed based on information from hillslope sediment budgets, image correlation techniques and 3D point cloud analyses. We pose that UAV-derived datasets provide essential data to improve our understanding of landslide kinematics and dynamics and of the hillslope soil residence time in landslide-prone catchments,
improving landslide hazard assessments. The UAV-SfM framework was applied to a study site in the Central European Alps of Switzerland where three distinct topographic datasets were generated for 2013, 2014 and 2015.

## 2 Method

### 2.1 Study area

The study area is the Schimbrig mass movement located in the northern foothills of the Central Swiss Alps, between Bern and Luzern (Figure 1). This mass movement is categorized as an earthflow, composed of centimetric to decimetric large clasts embedded in a matrix of silt and mud. The mass movement is located in the small, i.e. 4 km² large, Rossloch river catchment, which is part of the Entle river drainage system (Schlunegger et al., 2016a, 2016b). Previous studies on the Schimbrig landslide cover a broad temporal surveying scale from decades (Schwab et al., 2008) to one century (Savi et al., 2013). Our survey
complements earlier work in the sense that we will substantially improve the temporal and particularly the spatial resolution of monitoring. The catchment area is composed of three litho-tectonic units, oriented SW-NE. The upper part, including the Schimbrig ridge culminating at ca. 1,800 meters a.s.l., is formed by a Late Cretacous to Eocene suite of limestones, marls and quartzites, exposed in the Helvetic thrust nappes. The limestones form steep slopes, which are subject to rockfalls (Schlunegger et al., 2016b). Subalpine Flysch deposits are dominant in the intermediate part of the catchment between 1,100 and 1,400
meters a.s.l., where the earthflow is situated. This mass movement covers an area of ca. 45 ha, with a central active part where bare soil is exposed. The lower part of the catchment is covered by conglomerate bedrock knobs of the Subalpine Molasse,

which form small resistant conglomerates ridges and constrain the flow direction of the earthflow in its lower segment (Schlunegger et al., 2016b).

Seasonal slip rate variability of the area has been determined by Schwab et al. (2007) based on GPS measurements over a period of 14 months between 2004 and 2005. Slip rates were more intense in early spring and late summer and ranged between 0.1 to 0.25 m/month. During the other periods, slip rates were much lower and ranged between 0 and 0.1 m/month. This displacement pattern has been related to seasonality in soil moisture content. The same authors have not identified an immediate response of this earthflow to rainfall rates. At the decadal scale, volumetric changes of the Schimbrig flow were quantified using classic photogrammetry based on aerial photos from 1962 to 1998 by Schwab et al. (2008). They showed that extreme and episodic changes in slope morphology do not affect the long-term sediment transport to the channel network. Based on dendrogeomorphic analyses, Savi et al. (2013) qualitatively assessed the spatial pattern of geomorphic activity within the Schimbrig area over the last 150 years. Their analyses showed that the earthflow is a source of sediment, and the material mobilized by gravity is slowly supplying material to the drainage network by slow ground movements.

## 2.2 Data acquisition and data processing

In order to provide a spatial analysis of surface deformation through time, a time-series of very high resolution datasets of the Schimbrig area is necessary. Given the high geomorphic activity in the area (Schwab et al., 2008), yearly surveys were planned to accurately capture earthflow movements at high temporal resolution. Three image acquisition campaigns were organised in October 2013, June 2014 and October 2015. Late summer and late spring periods were chosen in order to minimize the effects of vegetation growth.

Image acquisition is done using a UAV platform, i.e. a custom Y6 multirotor with embedded DJI controllers. It is equipped with a stabilized camera mount, which compensates tilt and roll movements to maintain the fixed orientation of the camera. The used camera is a standard small format, i.e. 18.0 effective megapixels CMOS APS-C sensor, reflex camera (Canon EOS 550D). The lens is a fixed-focal-length lens (Canon EF 28mm f/2.8 IS USM), equipped with an optical image stabilizer and a high-speed autofocus motorization. The ground surface is surveyed both with nadir-oriented and with a 45°-tilted camera, as the inclusion of oblique images in the SfM algorithm has been shown to decrease systematic errors in topographic reconstruction and to better capture complex 3D structures (Clapuyt et al., 2016; James and Robson, 2014). Given a mean speed of 2 m/s to avoid motion blur induced by the UAV platform, the acquisition frequency is set at a rate of one image per two seconds in order to have a high overlap between consecutive images at ca. 60m above ground, i.e. above 70%, because high redundancy improves the performance of the SfM algorithm. Adjacent flight lines are separated by 15 meters, to maintain high lateral overlap of aerial images (Figure 2). Even though image acquisition has been performed in diverse light conditions depending on the hour of the day and cloud coverage during flights, the bottom line has always been to shoot in shutter speed priority mode and maintain a high value to avoid motion blur. Camera, flight and imaging parameters are summarized in Table 1 (O'Connor et al., 2017).

The point clouds are generated from overlapping optical images using the SfM algorithm. Structure-from-Motion is a computer vision algorithm that uses the set of unoriented overlapping pictures to reconstruct the 3D scene structure without additional a-priori information (Snavely et al., 2006, 2008). The output 3D point cloud is computed in a relative image-based coordinate system, and then georeferenced by matching the real-world coordinates of the georeferencing targets with those expressed in the image-based coordinate system. Based on the list of point pairs, the Helmert transformation parameters, i.e. translation vector, rotation matrix and scaling factor, are computed and applied to the entire point cloud. The output point clouds were georeferenced in post-processing based on ground control points (GCP), i.e. georeferencing targets, that were regularly scattered over the fly zone (Figure 2), and surveyed with a centimetric accuracy GPS receiver (Clapuyt et al., 2016). Despite the rough topography of the earthflow, a regular grid having 25 m distance between GCPs was possible. A limited number of measurements were not used in the final georeferencing as their associated error was abnormally high due to poor GPS signal reception. The georeferenced point clouds were filtered in order to statistically remove outliers. The root mean square error (RMSE) between the GPS coordinates and the position of the georeferencing targets in the point cloud serves as a measure of the accuracy of the 3D topographic reconstruction. These accuracy values are subsequently used to infer the error that is associated with the topographic representations and they define the detection limits for the multitemporal analyses. Finally, 3D point clouds are interpolated into digital surface models (DSM), which are then aligned. Both datasets serve as input for the analyses of the hillslope sediment budget and horizontal displacements. Resolution of the outputs was set based on the average density of 3D point clouds and standardized for all the datasets.

To quantify internal deformation of the Schimbrig earthflow, three distinct and complementary analyses are realized on the time series of 3D point clouds. First, the sediment budget is evaluated by computing DoD between DSMs, using the Geomorphic Change Detection software (Wheaton et al., 2010). Uncertainties associated to each surface representation, i.e. the accuracy values of 3D point clouds, are considered to be uniform over the entire study area. We subsequently analyse thresholded DSMs. The errors on the topographic representations are then propagated in the calculations of the DoD and quantities of erosion and deposition. By computing differences between DSMs, the mass balance of the earthflow is derived from the difference between the total eroded volume and the total deposited volume of sediments. Additionally, the average net thickness difference in elevation is computed by dividing the net volume difference by the surface area of the earthflow. Second, horizontal surface displacements are computed using the COSI-Corr image correlation algorithm (Ayoub et al., 2009; Leprince et al., 2007). Correlation is computed based on a moving window that scans dataset pairs. We used pairs of shaded relief surfaces to detect horizontal displacements. This method is particularly suitable for monitoring slow deformation processes, like the Schimbrig earthflow, with clearly distinguishable surface deformation structures, such as cracks, fissures and scarps. The COSI-Corr algorithm results into horizontal displacements expressed as two layers, i.e. the North-South component and the East-West component, which are combined to compute the intensity and direction of ground movements. The horizontal displacement analysis is complementary to the above-mentioned 3D analysis of point clouds: the presence of micro-topography and vegetation in the 3D point clouds facilitates the quantification of lateral earth movements using the

COSI-Corr algorithm. As such, the complexity of the earthflow including erosion and accumulation areas and horizontal displacements of the earthflow body is better represented. For each time interval, descriptive statistics, i.e. minimum, mean, median, quartile and maximum, are computed for the distribution of horizontal displacement values.

Third, 3D distances between point clouds are computed to measure the 3D topographic evolution through time, by highlighting zones of erosion, scarp retreat, surface subsidence and zones of bulging and sediment accumulation. The Multiscale Model to Model Cloud Comparison is used to compute 3D distances between point clouds because it directly operates on point clouds without meshing or gridding and provides a confidence interval associated to each distance measurement (Lague et al., 2013), in contrast to other cloud, mesh or raster distance computation techniques. For a set of core points, i.e. either the entire point cloud or a subsample, the method first computes surface normals in 3D. Then, along these normals, the local distance between clouds is the difference between the projection of core points on each cloud. Finally, a spatially variable confidence interval, i.e. a level of detection of local distance between clouds at 95%, is computed based on the registration error, i.e. the accuracy of point cloud georeferencing, and the local roughness of each core point projection along the normals. This confidence interval allows distinguishing statistically significant changes between two point clouds (Lague et al., 2013). Again, simple descriptive statistics are computed on the distribution of local distances between point clouds for each time interval.

The SfM processing and 3D point cloud georeferencing are performed using the Agisoft Photoscan® software. Sediment budgets were computed with the Geomorphic Change Detection software (Wheaton et al., 2010) implemented in ArcGIS. COSI-Corr algorithm has been used as a software module integrated in ENVI. Point cloud handling has been done with CloudCompare (CloudCompare version 2.6.3, 2016), using the M3C2 plugin. All other data manipulations were carried out with R software, CloudCompare and ArcGIS 10.

## 3 Results

### 3.1 3D topographic reconstructions

The three topographic reconstructions are not covering the same spatial domain (Table 2; Figure 3). The flight campaign of June 2014 allowed us to survey the entire Schimbrig earthflow. The 2013 and 2015 flight campaigns are centred on the most active part of the earthflow, but do not cover the entire earthflow due to operational problems encountered during the flight. Due to these spatial limitations, results are presented: (1) over the spatial intersection of the three datasets ($T_{2013} \cap T_{2014} \cap T_{2015}$), i.e. area of interest named *intersection*, to allow the comparison of absolute values of displacement and volumetric changes over the entire time period (2013-2015), and (2) over the spatial intersection of each time interval ($T_{2013} \cap T_{2014}$ and $T_{2014} \cap T_{2015}$), i.e. area of interest referred to as *interval*, in order to get the most information of each pair of datasets.

The 3D point cloud reconstructions of the Schimbrig earthflow result in a large dataset, with a very high point density of ca. 1,000 to 1,450 points per square meter, which allows to accurately track ground deformation through time. The three topographic reconstructions have similar accuracies, i.e. RMSE (Table 2), with horizontal accuracies of 0.23 and 0.20 m, and vertical accuracies of 0.06, 0.05 and 0.08 m. The total error on the topographic reconstructions ranges between 0.20 and 0.24

m. The overall detection limits for ground movements and deformations are derived from error propagation of the RMSE values on the individual topographic constructions given in Table 2.

The detection limits are similar for the two time intervals (Table 3), with a value of 0.31 m for 2013-2014 and 0.30 m for 2014-2015. For further analyses, changes which are smaller than the detection limit of our UAV-SfM framework are not reported.

## 3.2 Geomorphological map

Based on the time series of very-high resolution DSMs, their associated shaded relief surfaces, slope maps, and field observations, a geomorphological map (Figure 4) of the Schimbrig area is produced following Seijmonsbergen (2013) and the landslide classification by Varnes (1978). Field observations consist of GPS measurements of the scarp, flowlines and boundaries of the earthflow. Identification of stable ridges, and visual analysis of geomorphic units are based on expert knowledge.

The Schimbrig earthflow is constrained by stable bedrock ridges covered by trees on the northeastern part, and by relatively stable grasslands on the southwestern part. The main track of the earthflow is from the southeast to the northwest, and it contains two scarps in the upper part that expose up to 25 meters. The earthflow is characterized by a rough surface and a patchwork of vegetated surfaces covered by herbs and small shrubs and bare surfaces where wet and bare soil (silt and mud with embedded clasts) is exposed. The secondary track flows from the northeast to southwest and joins the main track upslope of its accumulation zone.

## 3.3 Sediment budget

Inputs for the sediment budget are the DSMs interpolated from 3D point clouds at the best spatial resolution possible, i.e. 0.04 m, based on the density of point clouds, after noise removal. To account for uncertainties related to the UAV-SfM reconstructions, a spatially uniform detection limit has been applied to each input dataset as defined in Table 2. Over the intersection of the surveyed areas, the estimated volume of deposited material is $6,012 \pm 1,919$ m³, for the first period of interest, i.e. between October 2013 and June 2014 (Table 4; Figure 5). The volume of eroded material is larger, with an estimate of $-11,345 \pm 3,118$ m³. For the first period, there is a net negative change in volume indicating that the removal of material is larger than the accumulation of sliding material for the surveyed surfaces. On average, there is lowering of the surface of $0.22 \pm 0.15$ m (Table 4). Field observations reveal a similar pattern. It is important to note that the lower part of the earthflow, where the debris is accumulating in a frontal lobe, is not included in the topographic analyses. Field observations indicate that only a minor part of the sliding material might have been transported to the river network via the Rossloch River.

The sediment budget computed over the second time interval, i.e. between June 2014 and October 2015, for the area of intersection is nearly at equilibrium (Table 4; Figure 5) with a net volume of difference of $-762$ m³ $\pm 3,450$ m³. This suggests that the sliding material that is mobilized by the earthflow accumulated within the area of intersection. When analysing the internal flow dynamics of the larger sliding area (interval) over the second time interval, it is clear that the sediment budget is

slightly positive (Table 4). This analysis now also captures the frontal lobe of the mass movement that bulged and advanced during this time interval. Figure 6 also shows that the upper and lower part of the earthflow are bulging areas, while the intermediate zone is experiencing surface subsidence.

### 3.4 Horizontal displacements

The COSI-Corr image correlation algorithm uses pairs of single-band input to quantify horizontal displacement. A north-directed illumination on the DSMs allows to highlight topographic features and ground deformation properly, as well as the presence of low vegetation. Shaded relief surfaces derived from the 3D point clouds (with a spatial resolution of 0.2 m) provided the best results for the correlation analyses. It is important to mention that input data with smaller spatial resolution generated incoherent displacement results, associated with high signal-to-noise ratio. In fact, under 0.2 m resolution, image

correlation led to false positive correlation between features due to the very complex topography of the earthflow. Pixels characterized by a signal-to-noise ratio lower than 0.9 or by displacement vectors that are smaller than the detection threshold are discarded for further analyses. To allow comparison between the two time intervals that have a different duration, i.e. 8 versus 17 months, the displacement values are here represented as mean annual displacement values (Table 5).

For the period between October 2013 and June 2014, the fluxes are relatively high and well confined by stable morphologic

ridges surrounding the earthflow (Figure 7). The displacement pattern is rather uniform over the entire length of the earthflow, suggesting that the earthflow body is moving downslope between stable bedrock ridges. The zones that show higher sliding activity are the lower scarp of the earthflow and its adjacent flat slope. The mean annual horizontal displacement for this first period is about 8.9 meters.

The output from the measurements during the second period of interest (2014-2015) shows a slightly different pattern of

surface displacements (Figure 8). The areas with highest displacements are located in the surroundings of the two scarps in the upper part, in the secondary track of the earthflow and in the lower accumulation zone were the frontal lobe spectacularly advanced downslope. In the flatter area between the two scarps, the horizontal movement is less pronounced and the direction of movement is rather diffuse.

When comparing the surface displacements for the two time periods over the same study area, it is clear that the magnitude of

the horizontal displacements is lower in the second period with a mean annual displacement of 5.7 m.

A comparison with field measurements realized during the flight campaigns in 2013, 2014 and 2015 indicates that the spatial pattern of the horizontal movements that were automatically extracted by the correlation algorithm are generally coherent with field observations: the stable morphologic ridges constrain the direction of the earthflow movement. Although the spatial pattern is highly consistent, the absolute displacement of the frontal lobe of the earthflow is not properly captured, as the frontal

lobe advanced by ca. 55 meters (Figure 9).

### 3.5 3D comparison of earth topography

Distances between 3D point clouds are computed using the M3C2 algorithm. For each pair of point cloud datasets, a subsample of the first 3D point cloud is taken as the set of core points, with a minimum distance of 0.5 m between points, to avoid extensive computation time. Descriptive statistics are computed on the point cloud distances (Table 6) which were filtered from values under the detection threshold.

The dominance of negative values can be interpreted as predominance of ground subsidence in flat parts or scarp retreat in steeper areas of the earthflow. Contrariwise, positive values are zones of surface bulging in the zones of accumulation or accumulation of debris at the frontal lobe.

## 4 Discussion

### 4.1 Application of UAV-SfM framework for landslide monitoring

Using the UAV-SfM framework, three very-high resolution topographic datasets were obtained over a 2-years period that allowed us to accurately quantify the internal dynamics of the earthflow and the sediment redistribution within the study area. First of all, the study confirms the efficiency of the UAV-SfM framework to perform natural hazard monitoring at very-high spatial resolution. In comparison to other high-resolution topographic methods (Passalacqua et al., 2015; Smith et al., 2015), the UAV-SfM technique, along with terrestrial laser scanning technology, provides the highest spatial resolution of surface reconstructions. Two main advantages of the UAV-SfM framework are its flexibility and its low cost and lightweight, which are convenient for repeated measurements of dynamic environments, and more particularly when deployed to acquire aerial imagery in remote and poorly accessible areas. Similar to terrestrial laser scanning that has similar spatial accuracy and range, the UAV-SfM framework requires careful setup and planning of georeferencing targets and image acquisition, to adequately and accurately capture the 3D scene of complex topography (Caduff et al., 2015; Clapuyt et al., 2016; James et al., 2017b). Being an airborne acquisition platform, the UAV-SfM has the advantage to configure a dense network of optimal viewpoints independent of terrain complexity and access difficulties.

The workflow applied for the research is based on our previous work on reproducibility of the technique (Clapuyt et al., 2016). The use of ground control points is time-consuming for large areas, as it requires a sufficient amount of observations that are well distributed around the study area. However, both criteria were fulfilled in our surveys. Only some GCPs, i.e. less than 5%, were discarded from the analysis because of the important associated error on the GPS measurement. Consequently, the use of the GCP georeferencing technique does not significantly affect our results, as observations were not included in the bundle adjustment step. However, we are confident that direct georeferencing technique for UAV-SFM framework will soon become a standard procedure as compact and low-cost RTK receivers are now available and suitable to be embedded on UAV platforms. Performance of direct georeferencing has recently been assessed and showed to be at least similar to GCP

georeferencing (Carbonneau and Dietrich, 2017; James et al., 2017a). In fact, this technique increases the field survey efficiency as the amount of GCPs can be significantly reduced and more importantly, it will increase the accuracy of topographic reconstructions where accessibility to place GCPs is drastically limited, e.g. volcanic terrains and glaciers. For further surveys of the Schimbrig earthflow, our workflow has to be adapted to include direct georeferencing. In this case study, a one-year time interval between surveys was suitable to capture the internal dynamics of the earthflow (Schwab et al., 2007). With respect to this, our 3D temporal database is considered as dense. We recognize that other natural hazards may require a higher frequency of measurements, which can easily be achieved using UAVs. Even though our results about landslide structure and dynamics need to be viewed against geophysical or geotechnical field investigations, this study shows the potential of time series of UAV-SfM topographic reconstructions to provide complementary data on internal landslide dynamics in complex terrain.

One of the main drawbacks of the UAV-SfM framework is the need for the UAV platform to acquire aerial pictures. The use of UAVs is now increasingly subjected to more stringent regulations, including a pilot license, UAV registration, and insurance certificates (Stöcker et al., 2017). Moreover, UAV flights are only possible under optimal meteorological conditions, and wind and rain may be a limiting factor in mountainous areas.

### 4.2 Schimbrig earthflow monitoring

Post-processing of the time-series of DSMs provides sediment budgets, horizontal displacements and 3D comparisons of surfaces that allow us to monitor the internal dynamics of the mass movement at very high spatial and temporal resolution. By qualitatively combining quantitative single results, which may seem redundant at first sight, it is possible to quantify the magnitude and rate of sediment redistribution and surface movements within the area affected by the Schimbrig earthflow and more importantly to capture the internal mechanisms of the earthflow (Figure 10).

As such, the very-high resolution topographic reconstructions allow to analyse the spatio-temporal evolution of earthflow-prone terrain, and to go beyond conventional survey methods and expert knowledge (Savi et al., 2013; Schwab et al., 2007, 2008). Savi et al. (2013) provides a historical insight of the Schimbrig sliding activity based on dendrochronology over the last 150 years. These qualitative data on hillslope processes provide information on surface displacements that extends beyond the timespan of photogrammetric techniques, but the temporal information has its limits regarding quantitative and continuous data on displacement rates. A first attempt to quantify slip rates of the earthflow was carried out by Schwab et al. (2007) who tracked control points using GPS measurements. However, we are now able to compute slip rates over the entire extent of the earthflow at very high temporal resolution applying image correlation algorithms to very high-resolution aerial images. Schwab et al. (2008) complemented their topographic analysis of the earthflow using classic photogrammetry and aerial photographs spanning the last 50 years. Independent of the spatial resolution of the output and price of data acquisition, photogrammetric analyses are very valuable as they allow to capture surface displacements at larger scales and over longer time periods than

data derived from the UAV-SfM framework, even though time interval between the data sets may be a limiting factor (e.g. Prokešová et al., 2010; van Westen and Lulie Getahun, 2003). Schwab et al. (2008) emphasized that sediment fluxes in the trunk streams are not directly controlled by the production of loose material through landsliding on the hillslopes. This is confirmed by our high-resolution topographic analysis, and the computed hillslope sediment budget.

The UAV-SfM framework allows to go a step further in the interpretation of the earthflow's internal structure and dynamics.

When superimposing the temporal series of the three sets of topographic analyses and interpreting it in very detailed way, we are able to show the complex rotational structure of the earthflow (Figure 10). Our data show that the entire body of the earthflow is sliding, but that there exist strong differences in internal deformation and flow velocities within the sliding
material. By combining the results from the 3D comparison between the point clouds and the DoD, it is possible to map the succession of ground surface subsidence and bulging areas over the three-year period. Notwithstanding the short monitoring interval, the pattern of internal deformation of the earthflow changed its configuration (Figure 10). Between October 2013 and June 2014, the earthflow had a succession of three nested rotational units. In the upper part of the study area, two steep scarps are present. These active scarps control the downward movement of two tilted blocks (see Figure 4). The location of the two
main scarps advances by ca. 8 m during the 2013-2014 period. A third rotational unit is larger: it is defined by a steep scarp located in the middle part of the earthflow, and extends over the lower and flatter part of the earthflow down the frontal lobe. During the second period of monitoring, i.e. between June 2014 and October 2015, two rotational units can clearly be distinguished, i.e. a small upper block that is confined above the upper active scarp, and a larger heterogeneous sliding mass that extends down to the frontal lobe of the earthflow. Unlike the first period of interest, the lower erosional scarp is not active,
and there is no distinction between the second and third part of the sliding mass.
The very-high resolution spatio-temporal analyses demonstrate that the Schimbrig earthflow has been very active over the monitoring period. Results from the image correlation algorithm highlight the strong internal redistribution of sliding material within the earthflow, and rapid changes in the spatial pattern of displacement vectors. The mean annual horizontal displacements are large with values of ca. 9 m between October 2013 and June 2014 and ca. 6 m between June 2014 and
October 2015. This is partly explained by the fact that the central part of the earthflow is advancing toward the foot of the earthflow between October 2013 and June 2014, and this advance is accompanied by surface subsidence along the main track. This phase is followed by bulging in the accumulation zone, and a strong advance of the frontal lobe of the landslide over a distance of ca. 55 m.

Notwithstanding the strong internal deformation of the sliding material, there is no net effect on the sediment flux at the outlet of the Rossloch River. Our data show that the overall sediment budget of the earthflow is nearly in equilibrium and is spatially very consistent with the results on the displacement vectors and distances derived from the point clouds. After the major surge that occurred in 1994, the earth surface lowered by ca. 12 m in the central track of the earthflow (Schwab et al., 2008). More than 20 years later, the earthflow shows strong internal deformation that is related to the re-adjustment and self-reorganisation

of the sliding material after the 1994 surge event. This suggests that phases of enhanced earthflow kinematics are not necessarily leading to enhanced sediment export to the fluvial system, because of the time delay between successive phases of earthflow reactivation and the sediment export from the catchment.

## 5 Conclusion

The UAV-SfM framework is increasingly applied in geomorphology to accurately capture the topography of given scenes. As it is low cost and flexible in its implementation, and particularly suitable for surveying dynamic environments in poorly accessible terrain, we used this methodology to quantify ground surface displacements of the Schimbrig earthflow, located at the foothills of the Central Swiss Alps, at very-high spatial and temporal resolution. Based on three topographic reconstructions between autumn 2013 and 2015, we were able to conduct a comprehensive 3D analysis of the landslide by combining the

sediment budget of the hillslope, and the horizontal and the 3-dimensional surface displacements.

Combination and careful interpretation of the three topographic analyses of this study allow us to reconstruct the internal dynamics of the earthflow and highlight its complex rotational movement. In addition to variation in horizontal surface movements through time, we interestingly showed that the rotational structure of the earthflow is also varying from year to year. Additional field surveys will be required to increase the temporal series of topography reconstructions and confirm our

findings. Results also confirm that the Schimbrig earthflow is very active with mean annual horizontal displacements between 6 and 9 m. Besides, we showed that the sediment budget of the hillslope is nearly at equilibrium. In fact, the earthflow has experienced a major sediment pulse more than 20 years ago, and is still re-adjusting to this new setting. Therefore, our very-high spatial and temporal resolution analysis supports the findings of Schwab et al. (2008) about the time lag between sediment production on hillslopes and fluvial processes, i.e. enhanced sediment fluxes, in trunk streams, based on lower spatial and

temporal resolution time series of DEMs. Finally, this short-term monitoring of the Schimbrig earthflow nicely stacks up on the previous studies done over the same study area.  It brings an additional value to allow further research, which will integrate all these spatial and temporal scales in term of erosion and hillslope processes.

**Competing interests**

The authors declare that they have no conflict of interest.

**Acknowledgements**

Authors would like to thank Emilien Aldana Jague and Marco Bravin for the help and the support during test flights with the
UAVs used to acquire aerial images. We would also like to acknowledge Romain Delunel from Bern University for its
availability and help regarding the handling of the GPS used to survey ground control points during field work. Furthermore,
authors thank two anonymous reviewers and the handling associate editor, Anette Eltner for their constructive comments.
Low-cost research is also possible thanks to the developers making their software freely available: Dimitri Lague and Nicolas
Brodu for M3C2 algorithm; the COSI-Corr development team, from Caltech (California Institute of Technology) Tectonics
Observatory; Joe Wheaton and colleagues for the Geomorphic Change Detection software.

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

**Figures**

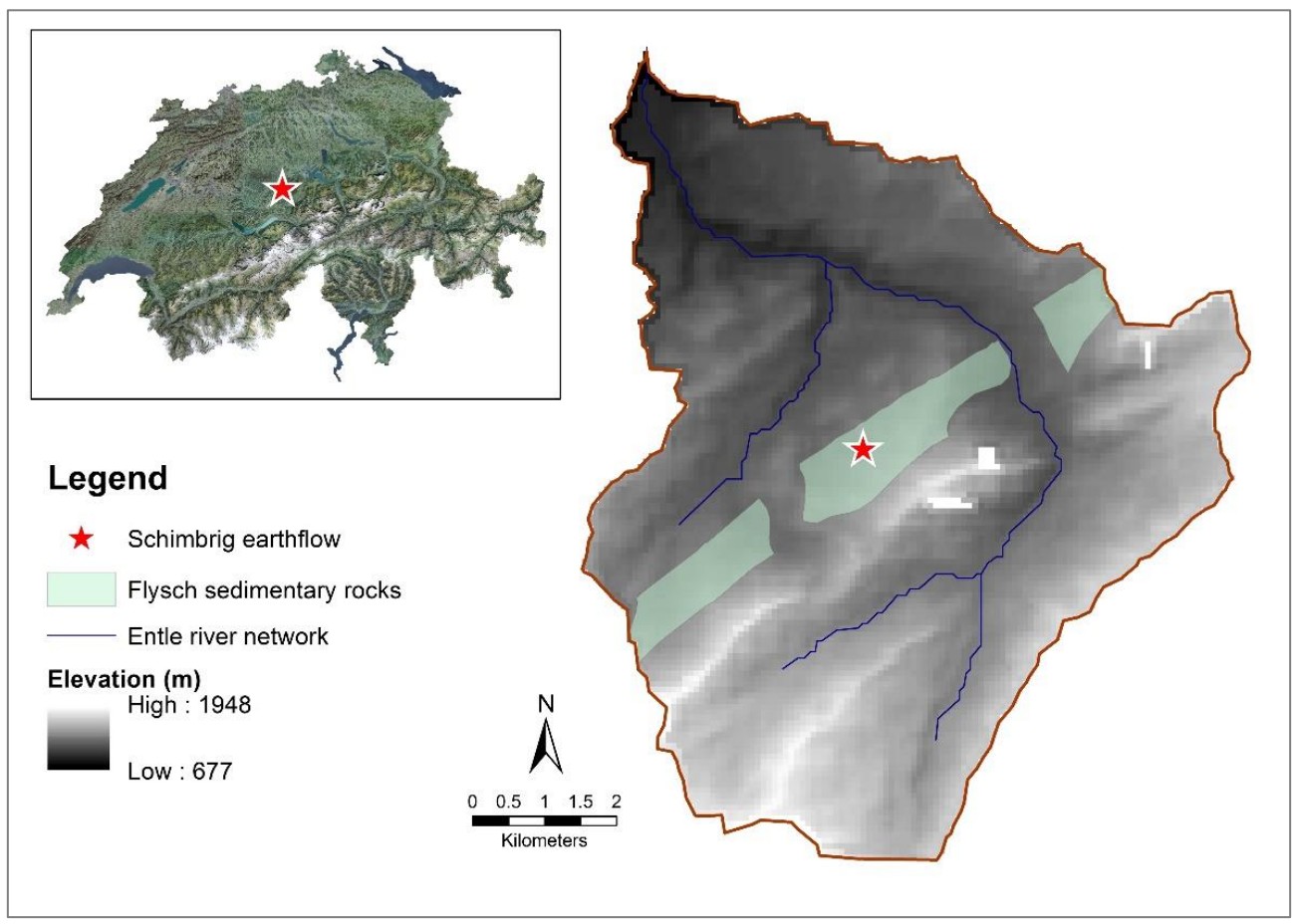

**Figure 1: Location of the Schimbrig landslide in Switzerland (inset) and in the Entle river watershed.**

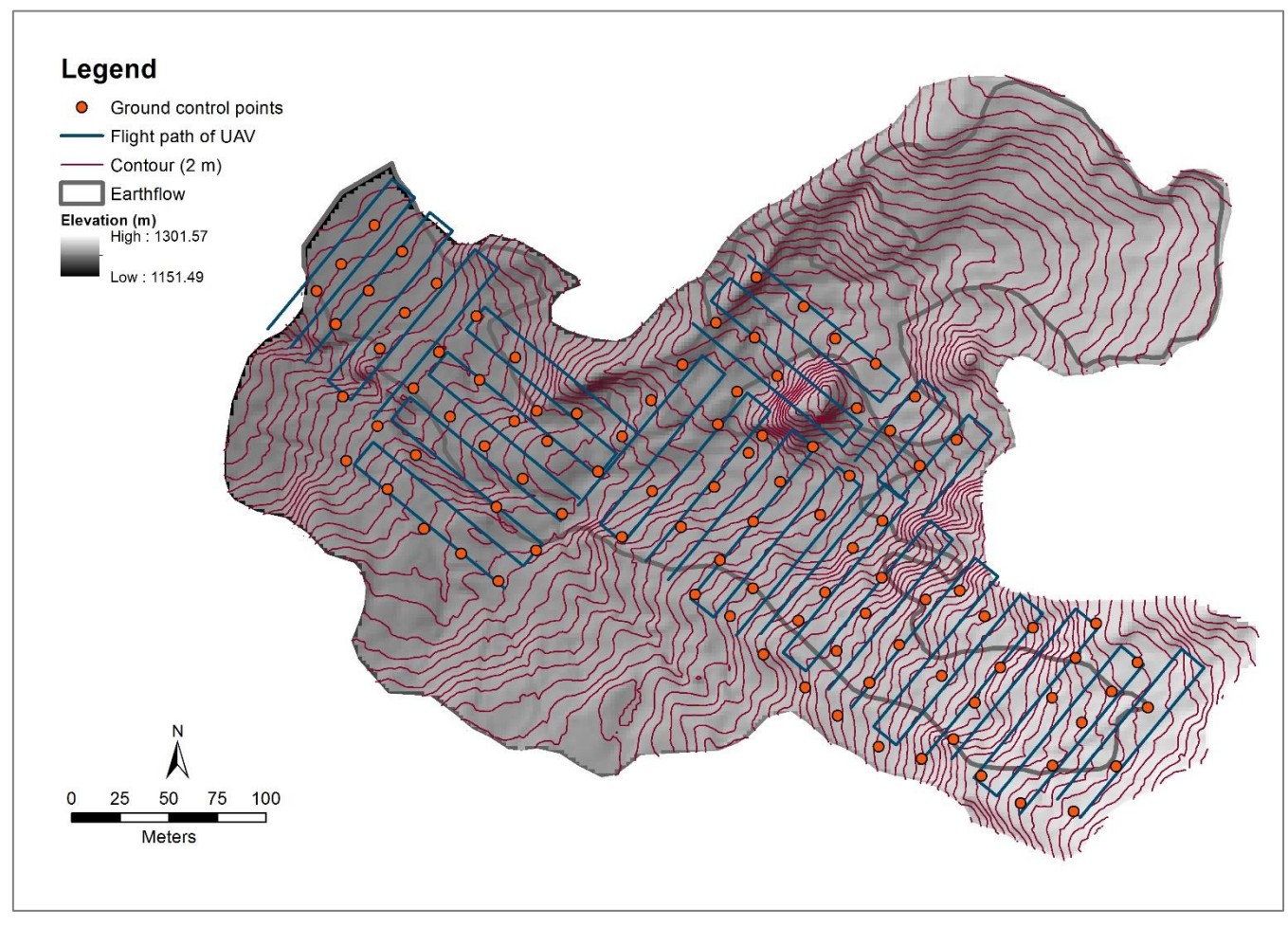

**Figure 2: Settings of the UAV flights and GCPs location (example from the October 2015 survey).**

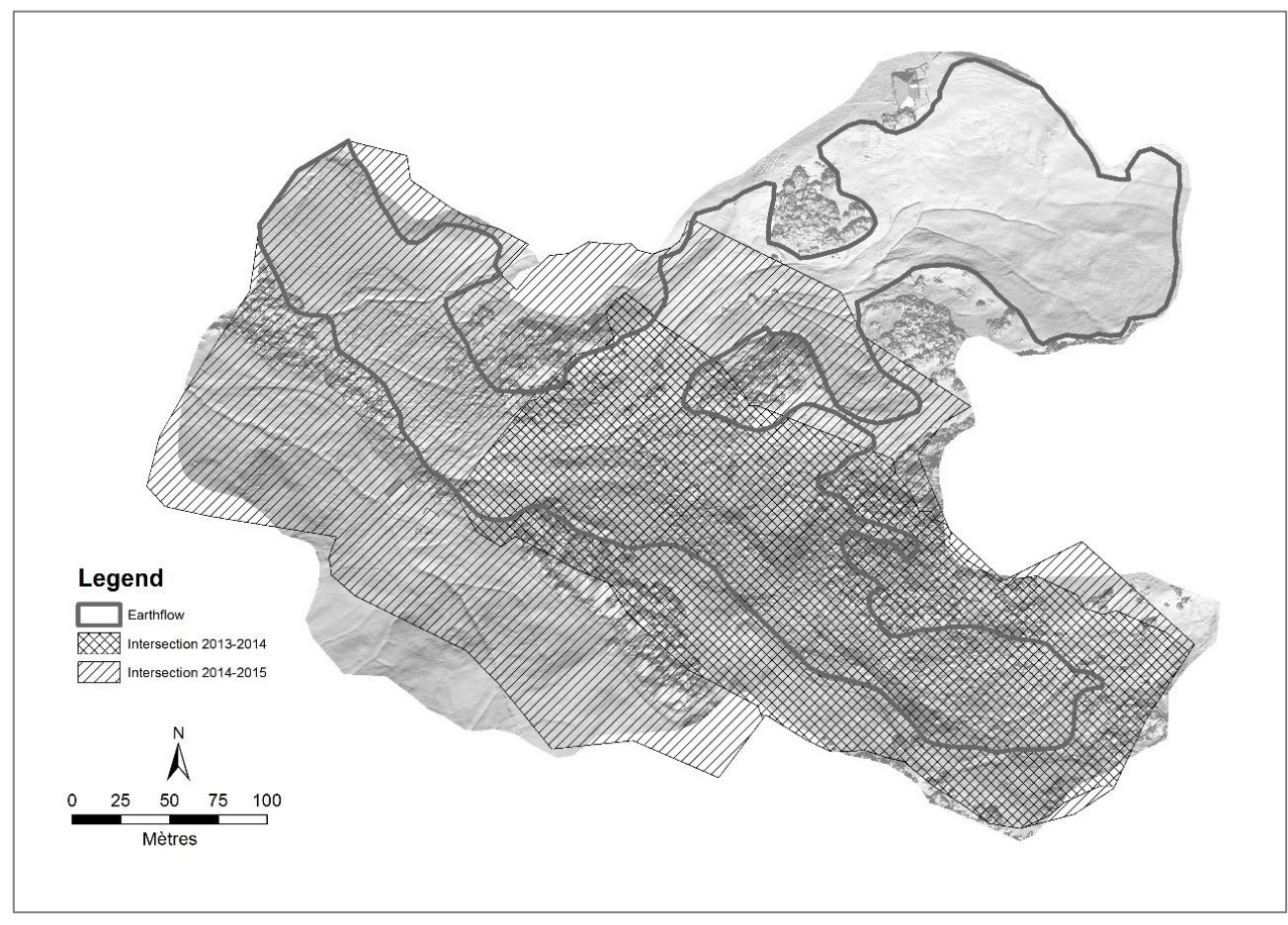

**Figure 3: Intersections of point cloud datasets for the 2013-2014 and 2014-2015 time intervals.**

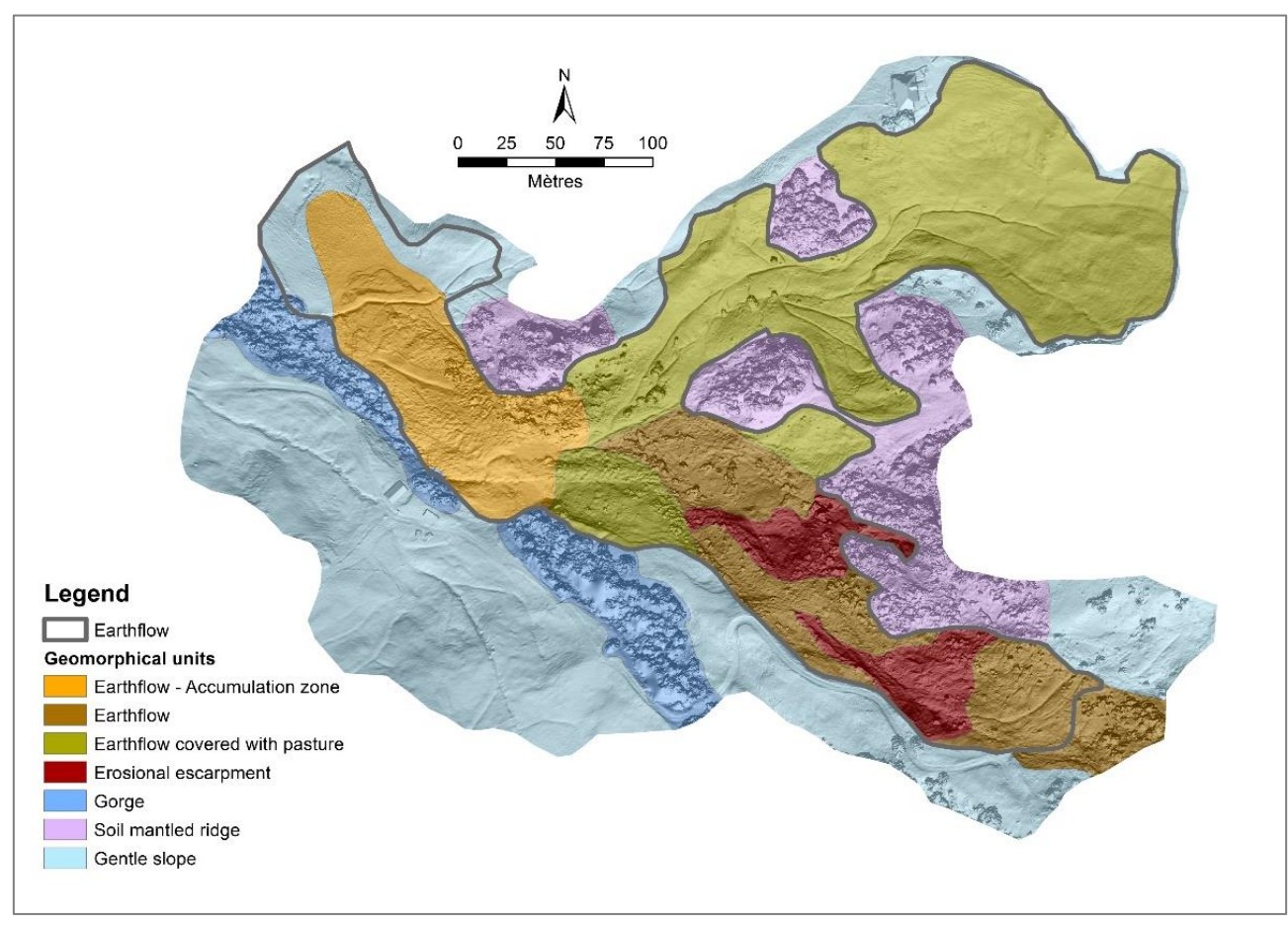

**Figure 4: Geomorphological map of the study area.**

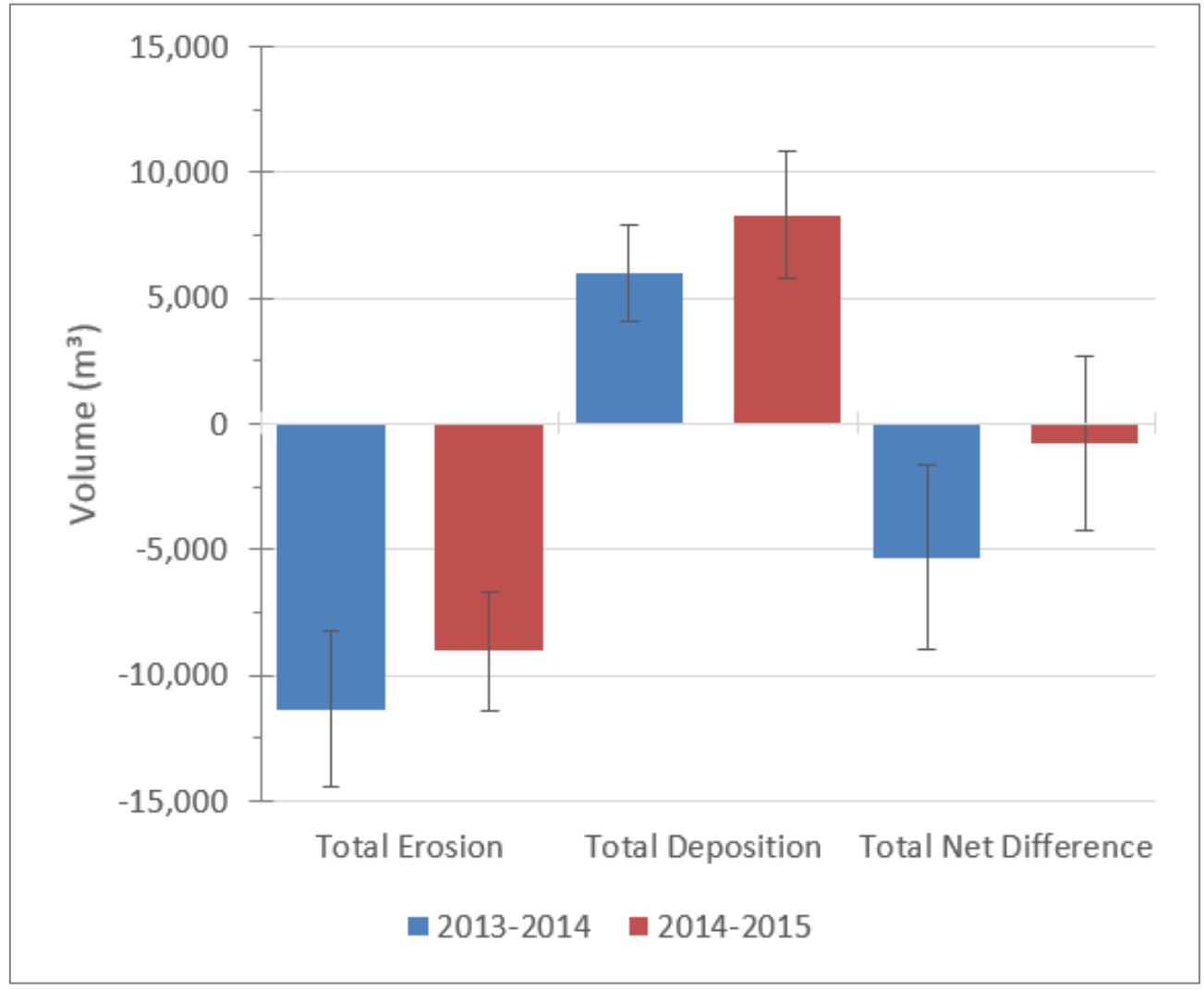

**Figure 5: Volumetric sediment budget computed for the spatial intersection of the three datasets (see Table 4 for details).**

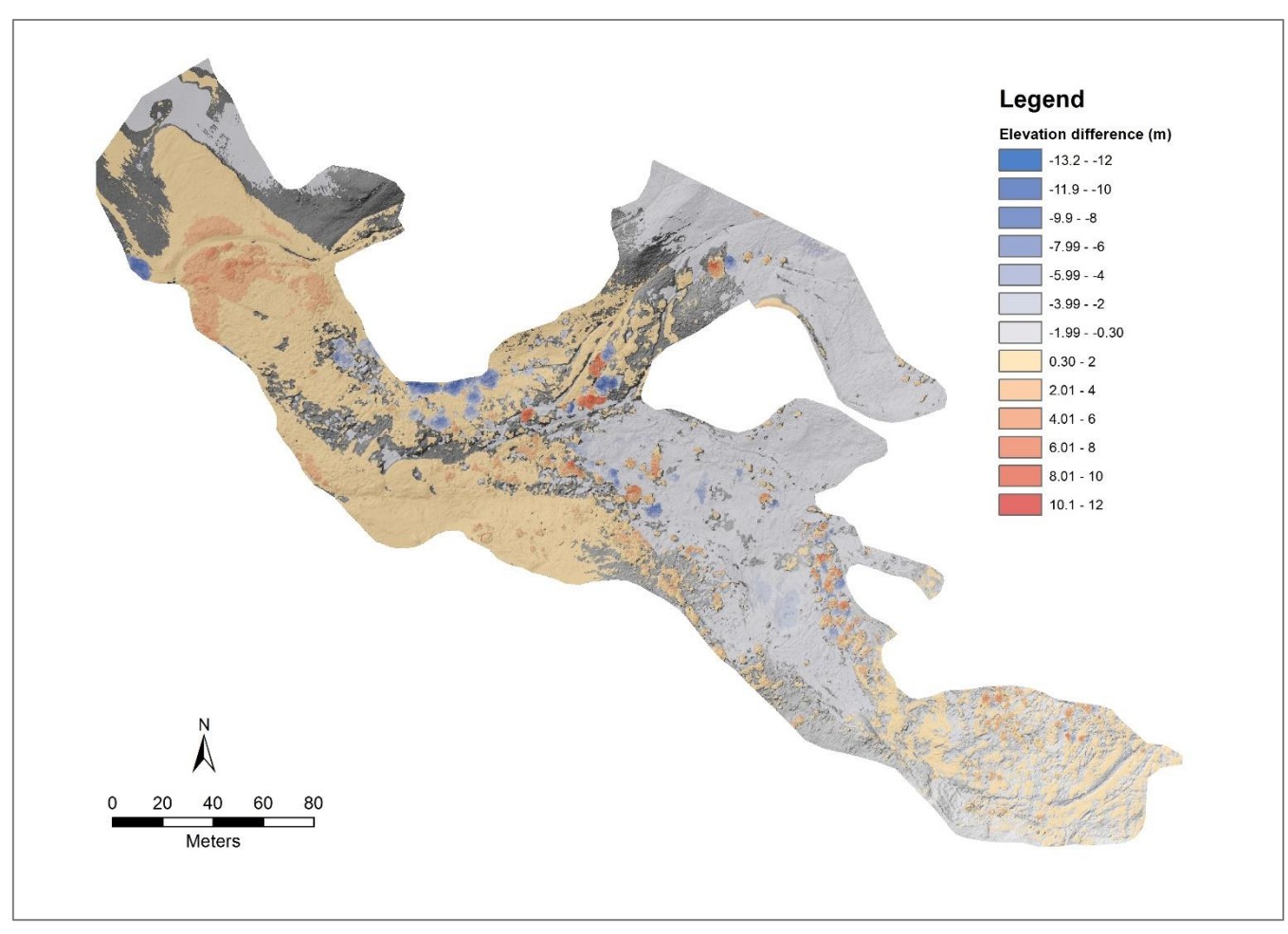

**Figure 6: Elevation difference between two topographic reconstructions for the time interval June 2014 - October 2015.**

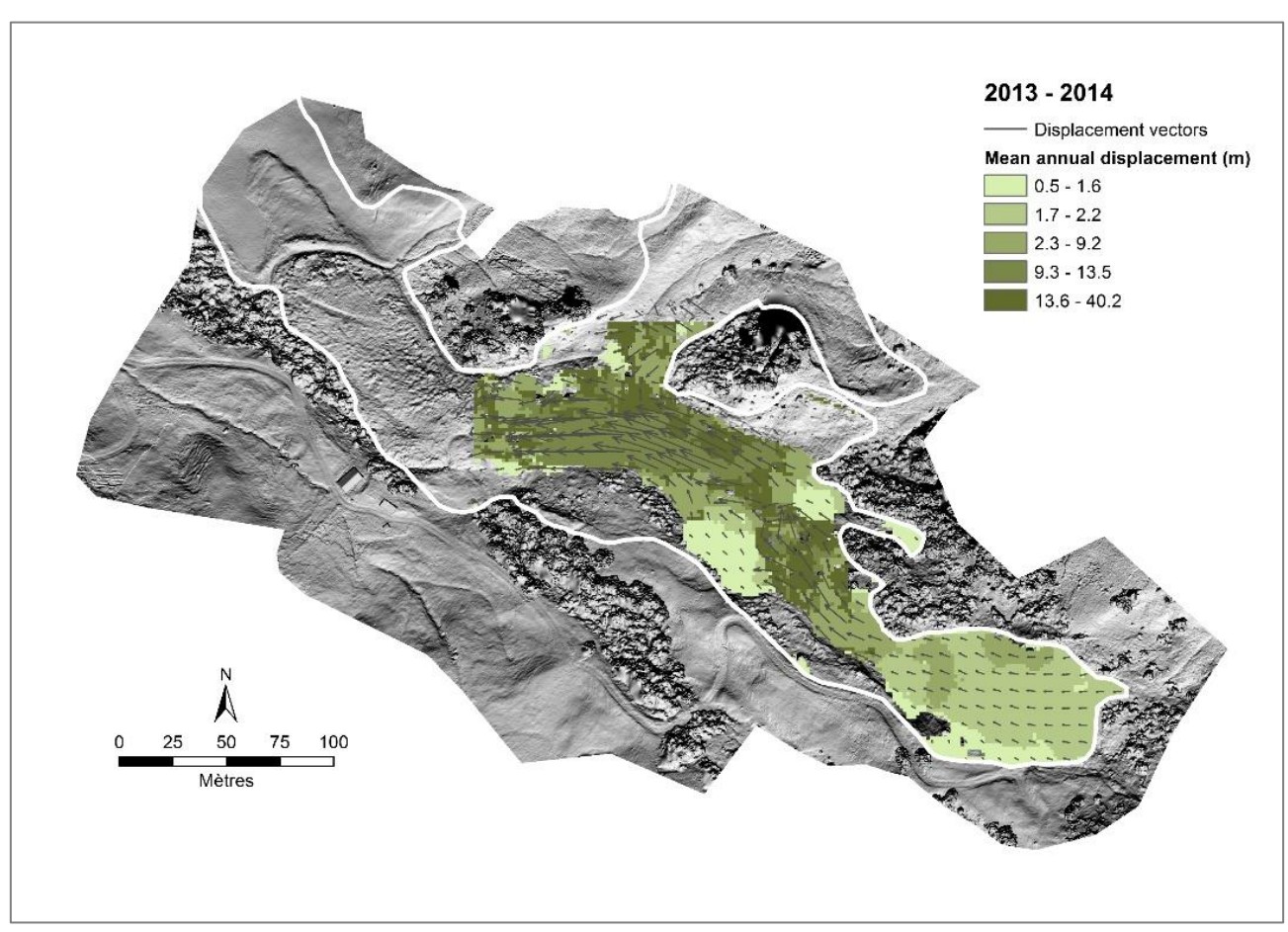

**Figure 7: Surface displacements for 2013-2014 using the COSI-Corr algorithm. Displacement vectors are indicative of relative magnitude and direction of movement and do not reflect true ground displacements.**

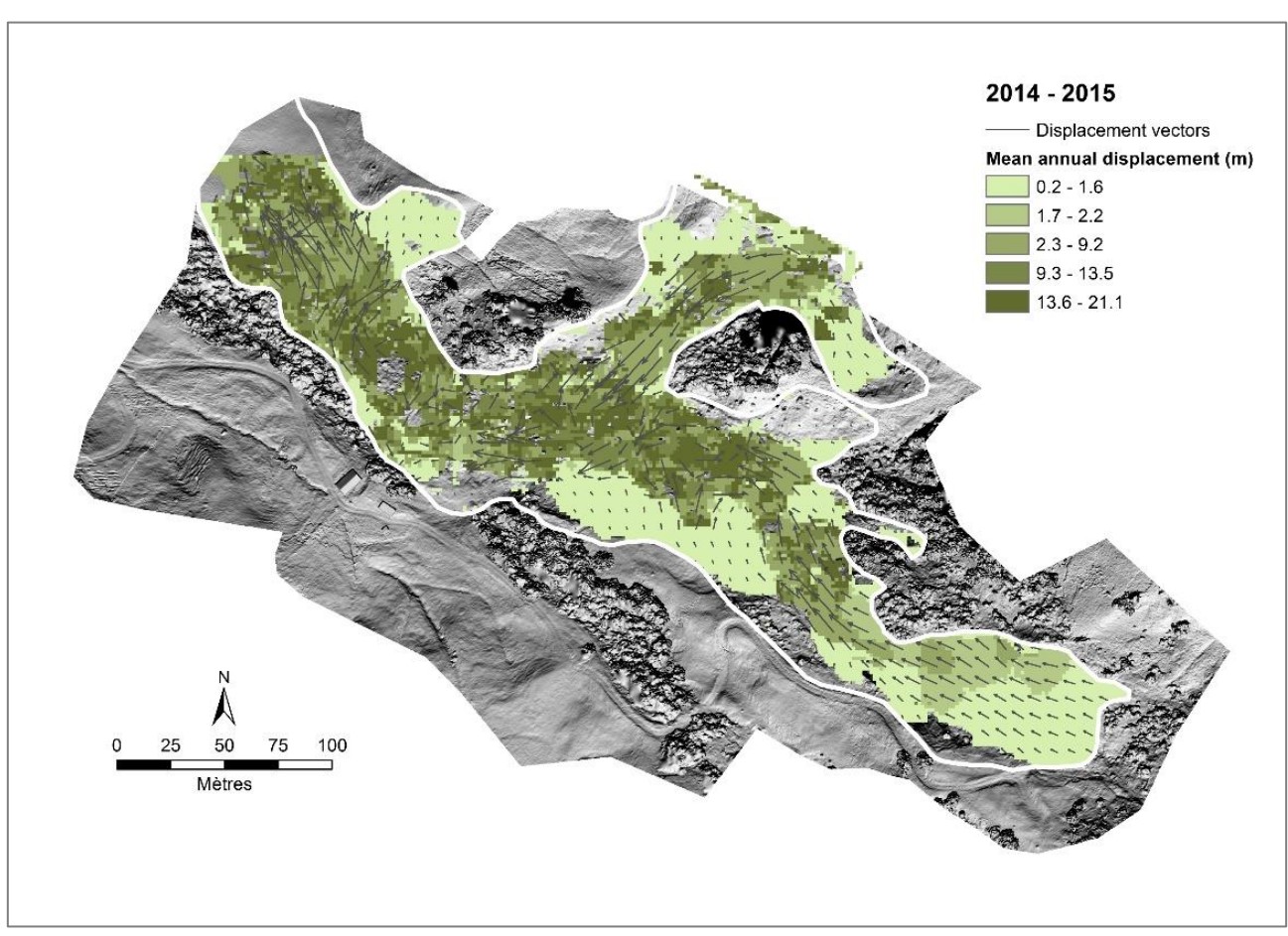

**Figure 8: Surface displacements for 2014-2015 using the COSI-Corr algorithm. Displacement vectors are indicative of relative magnitude and direction of movement and do not reflect true ground displacements.**

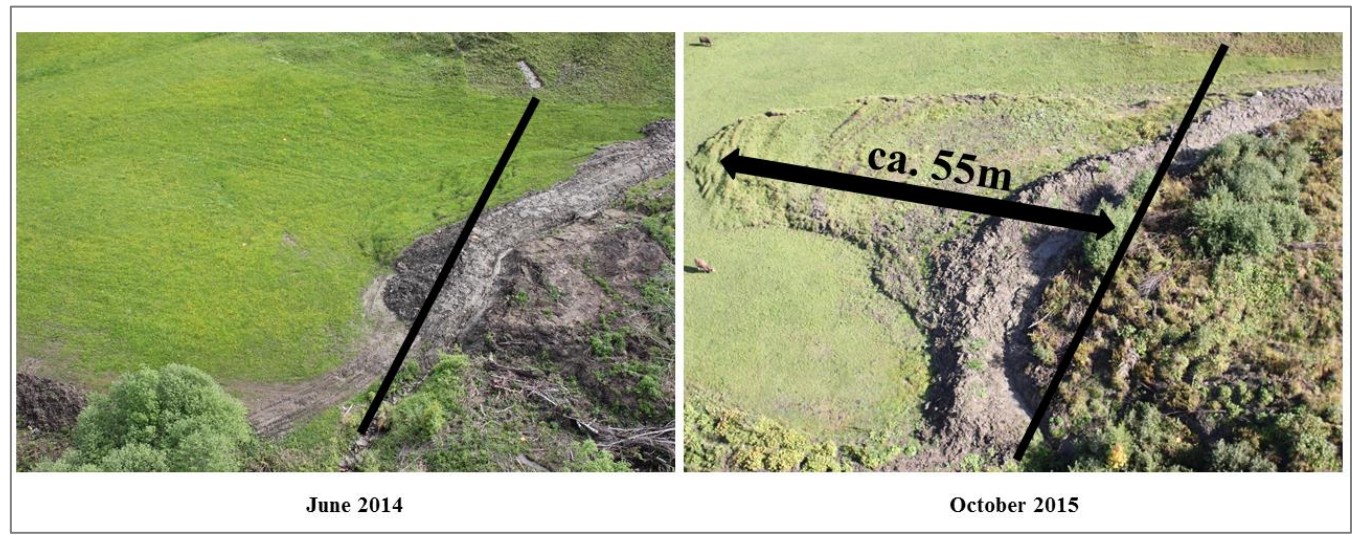

June 2014                                    October 2015

**Figure 9: Aerial images illustrating the advance of the frontal lobe of the earthflow between June 2014 and October 2015.**

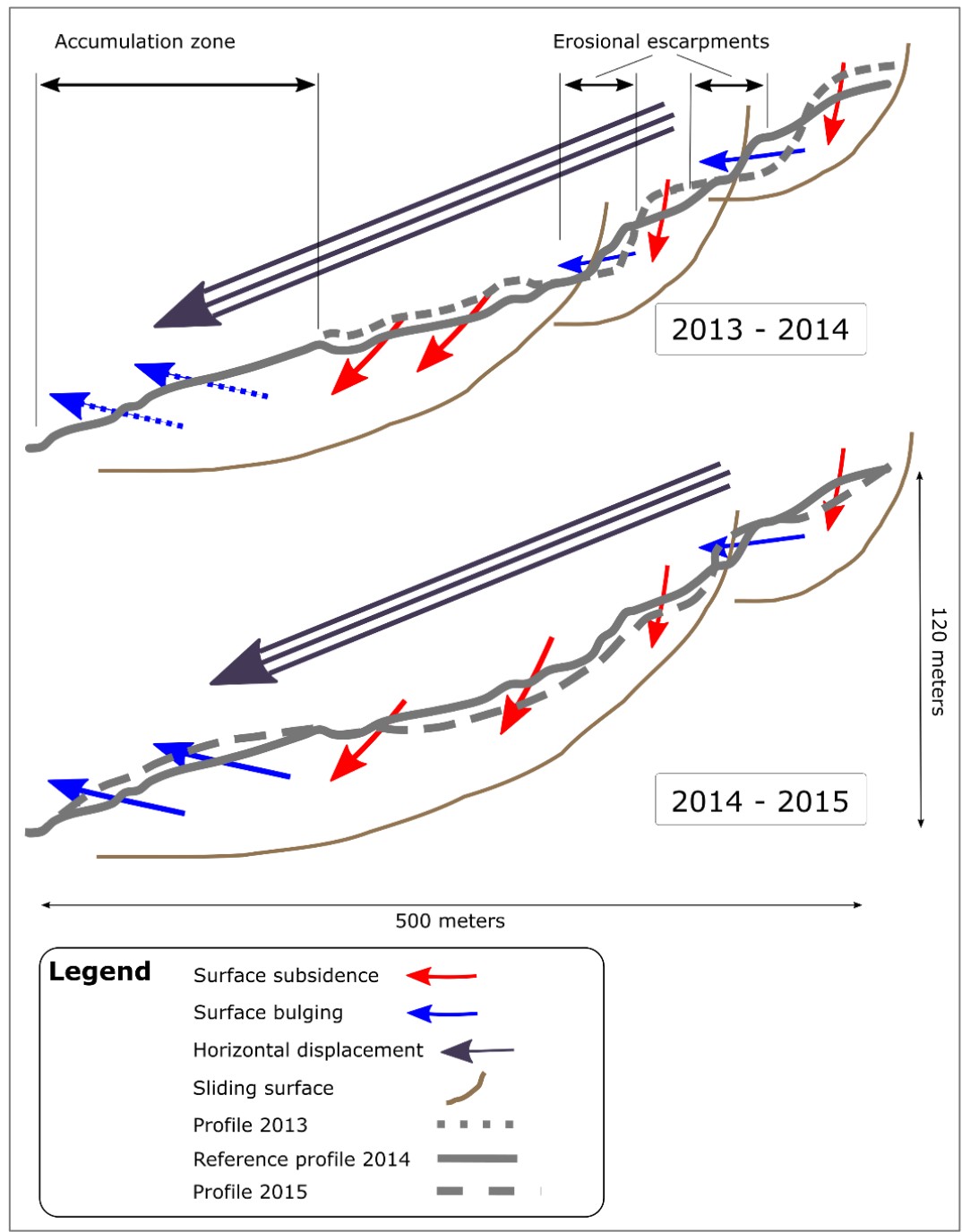

**Figure 10: Schematic summary of the internal dynamics of the Schimbrig earthflow based on very-high resolution monitoring using the UAV-SfM framework.**

**Tables**

Table 1: Camera and imaging characteristics for all the surveys.

| | |
|---|---|
| **Camera Model** | Canon EOS 550D |
| **Lens Model** | 28mm f/2.8 IS USM |
| **Image resolution** | 5184 * 3456 pixels |
| **Crop factor** | 1.6 |
| **Approximate sensor size** | 22.3 * 14.9 mm |
| **Pixel pitch** | 4.3 µm |
| **Mean shutter speed** | 1/1250 |
| **Mean ISO** | 388 |
| **Mean f-number** | 3.49 |
| **Flight velocity** | 2 m/s (idealised) |
| **Flight height** | 60 m (idealised) |
| **Ground sample distance** | 9.2 mm |

Table 2: Characteristics and accuracy assessment of 3D point cloud reconstructions. Root mean square errors (RMSE) are the standard deviation of differences between the coordinates of the ground control points, i.e. georeferencing targets, measured by GPS and the coordinates of these points within the 3D point clouds after georeferencing. Horizontal RMSE is computed by taking the horizontal components of the point coordinates while the vertical RMSE is computed by taking only the z component of point coordinates.

| | | October 2013 | June 2014 | October 2015 |
|---|---|---|---|---|
| **Point cloud characteristics** | **Nb of pictures** | 1143 | 4519 | 2501 |
| | **Nb of GCPs** | 49 | 108 | 99 |
| | **Area (ha)** | 5.86 | 17.69 | 15.59 |
| | **Point density (pts/m²)** | **1456** | **1270** | **1080** |
| **RMSE (m)** | **Horizontal** | 0.23 | 0.20 | 0.20 |
| | **Vertical** | 0.06 | 0.05 | 0.08 |
| | **Total error** | **0.24** | **0.20** | **0.22** |

**Table 3: Limit of detection values for temporal analyses (meters).**

|  |  | 2013 - 2014 | 2014 - 2015 |
|---|---|---|---|
| **Limit of Detection (m)** | **Horizontal** | 0.30 | 0.28 |
|  | **Vertical** | 0.08 | 0.09 |
|  | **Total** | **0.31** | **0.30** |

**Table 4: Volumetric (cubic meters) and depth (meters) sediment budgets.**

| Area of interest | Intersection | | | | Interval | |
|---|---|---|---|---|---|---|
| **Time interval** | **2013-2014** | | **2014-2015** | | **2014-2015** | |
|  | **Estimate** | **Error (±)** | **Estimate** | **Error (±)** | **Estimate** | **Error (±)** |
| **Total Volume of Erosion (m³)** | -11,345 | 3,118 | -9,054 | 2,329 | -15,093 | 4,098 |
| **Total Volume of Deposition (m³)** | 6,012 | 1,919 | 8,293 | 2,546 | 17,005 | 4,533 |
| **Total Net Volume of Difference (m³)** | -5,333 | 3,661 | -762 | 3,450 | 1,912 | 6,111 |
| **Average net thickness difference (m)** | -0.22 | 0.15 | -0.03 | 0.14 | 0.05 | 0.15 |

**Table 5: Mean annual horizontal displacement (meters) computed using the COSI-Corr algorithm applied on shaded relief surfaces.**

| Area of interest | Intersection | | Interval |
|---|---|---|---|
| **Time interval** | **2013-2014** | **2014-2015** | **2014-2015** |
| **Min.** | 0.47 | 0.23 | 0.23 |
| **1st Qu.** | 1.81 | 0.89 | 0.89 |
| **Median** | 3.74 | 2.20 | 3.62 |
| **Mean** | 8.91 | 5.74 | 6.30 |
| **3rd Qu.** | 13.06 | 11.25 | 11.84 |
| **Max.** | 40.16 | 21.06 | 21.06 |

**Table 6: Descriptive statistics of the distances between 3D point clouds (meters).**

| Area of interest | Intersection | | Interval |
|---|---|---|---|
| **Time interval** | **2013-2014** | **2014-2015** | **2014-2015** |
| **Min.** | -20.72 | -27.56 | -28.83 |
| **1st Qu.** | -1.06 | -1.00 | -0.89 |
| **Median** | -0.60 | -0.49 | -0.43 |
| **Mean** | -1.04 | -0.69 | -0.38 |
| **3rd Qu.** | 0.47 | 0.66 | 0.79 |
| **Max.** | 20.36 | 26.96 | 26.96 |