# Peer review of "Unravelling earth flow dynamics with 3D time series derived from UAV-SfM models"

_Earth Surface Dynamics, 2017_

## Referee Comment (RC2)

In the manuscript, Clapuyt et al. present the results of UAV photogrammetric surveys carried out on an active landslide in Switzerland. Their results have been used to quantify the horizontal and the 3D ground surface displacements and the sediment budget of the landslide. The Authors focused the manuscript on the interpretation of the landslides dynamics based on the high resolution dataset provided by UAV photogrammetry and they used M3C2, COSI-Corr and GCD (ArcGis Plugin) to obtain a comprehensive analysis of the annual dynamic of the landslide.

The manuscript is well written and the work is very interesting and potentially useful for future developments of UAV photogrammetry for landslide monitoring. However, there are some points of the paper that require improvements. In my opinion, the manuscript require a minor revision before being considering for publication in ESurf. I include below some suggestions or comments that could be of interest for the authors to be incorporated in the final version of the manuscript.

**General comments/suggestions**

1) Accuracy has been assessed comparing SfM photogrammetry with the ground control points used to georeferenced the dense point cloud. However, as visible in Figure 2, both surveys, 2013 and 2015, were able to cover some area outside the earthflow. In addition you used the digital elevation model (DEM) and the elevation difference on the common area to estimate the sediment budget. Why do not consider also the elevation difference between multi-temporal DEM on the stable areas outside the earthflow as additional analysis to evaluate the accuracy of the SfM reconstruction? This could be useful to evaluate the spatial distribution of elevation changes between the three survey campaigns.

2) In my opinion, the Authors not emphasize the advantages of very high resolution UAV data (both point cloud and DEM) for the landslide monitoring in comparison with the state of the art and previous investigations done by Scwab et al. (2007) and Savi et al. (2013) on the same study area. In addition, I suggest to better highlight how the results derived by the three different methodologies can be combined and which improvement their combination can provide on the interpretation of the landslide dynamic.

3) Concerning the structure of the paper I have some observations, starting from the introduction where, in my opinion, some information are missing and I found it a bit confused.
*Introduction*. The Authors report a general description of the high resolution techniques available for the reconstruction of earth's landform. Then, a sentence about the accuracy is provided, following by a more detailed description about spatial resolution and spatial extension for each sensor and platform. In the second paragraph, the concept of high resolution is repeat again regarding the landslides monitoring and surface displacements. In my opinion these two paragraphs, should be rewritten focusing on the target object, i.e. landslide monitoring, by giving a clear description of the advantages and disadvantages of different survey technologies currently used for landslides monitoring and surface displacements analysis. In the introduction, two times you wrote about the aspects that affect the choice of the technologies. In specific, at line 10 you mention that "the choice of the acquisition framework result from the trade-off between the spatial resolution needed and the extent of the study area", then at line 20 you mention that "return period for acquisition and the surveying cost remain important criteria for the selection of the data acquisition platforms". These two aspects ( i.e. return period and cost)also affects the acquisition framework. I assume that the resolution and return period for acquisition necessary for landslide monitoring are strongly site-specific and depends by the magnitude and the assessment of associated hazard. However, I suggest to consider to write which are the main parameters (like resolution, data type, weather, accuracy, location accessibility, spatial and temporal resolution, coverage, cost) to consider when making a choice between different high resolution technologies, focusing on the landslide monitoring. Since you mentioned in the text different technologies, please consider that in the last decade both satellite (i.e.

very high resolution satellite imaging) and aerial imaging system have benefited from great technology improvements reaching similar sub-meter resolution. Recently Stumpf et al. (2017) investigated the potential of Pléiades satellite images for landslide monitoring. I suggest to describe the real advantages of UAV data like the 3D point cloud, cm resolution, etc. Maybe, I suggest to refer here the comparison of your study with the previous investigations done by Schwab et al.(2007), and Savi et al.(2103).

**Specific comments/suggestions**

P4, line 8. I consider inappropriate to add a sentence about the effect of climate change on landslide hazard at the end of the introduction and after the description of your work. Maybe consider to start the introduction with this general topic that help to focus the object of the manuscript.

P4, line 2. First you computed the 3-dimensional surface displacement, then the horizontal and the sediment budget. Please change the order. In addition I suggest to introduce the acronym 3D at the beginning (P3, line 14) and use it in the entire text.

If you consider to extend the DoD on the stable area in order to analysis the accuracy of the photogrammetric DEM, I suggest to firstly describe the sediment budget based on the DoD, then the COSI- Corr analysis and at the end the M3C2. This also because for the COSI-Corr the hillshaded DEM is the data input (P6, line17). Moreover, the DoD provides information mainly about the vertical change, COSI-Corr the horizontal displacements and M3C2 is a full 3D analysis. I believe this sequence of the analyses more appropriate. If you change the order, then you should verify that you change it throughout the whole paper.

P5, line 11. Consider to change: Data acquisition and data processing.

P5, lines 12-13. Is it really necessary "a 3D point cloud" or this is a sentence related to your study.

P5, line 14. The acronym SfM is already introduced. In addition I cannot find the connection of this sentence where you introduce the SfM algorithms with the next one about the planning of the survey. Please clarify this sentence and consider to move it at line 28 when you introduce the SfM algorithms.

P5, line18. Please provide more information about the platform, data acquisition and processing like the type of the UAV platform, flight height and flight path, GSD for each epoch, number of oriented images and very important the locations of the GCPs on the survey area. If I understood well, the GCPs were used after the camera orientation to georeference the point cloud. Why you didn't consider to include some of these observations in the bundle block adjustment during the camera orientation and the remaining as check points? Perhaps worth a comment.
Please consider to include in this section also the description of the DEM generation and the accuracy used for the different analysis. You explained that for the horizontal displacement a spatial resolution higher than 0.20 generated incoherent results. However, for the DoD you used 0.04 m cell size as the best possible spatial resolution. How did you estimated this, based on the GSD? Why not used the same cell size, considering that the mean annual horizontal displacement range between 5.7 m to a max. of 8.9 m? Usually photogrammetric point cloud is characterized by high noise. Did you remove the noise before to generate 4 cm resolution DEM.

P6, line 4. Consider to change the order of the analyses.

P6, line 29. In order to avoid repetition, better to report the corresponding software in the specific paragraph. Please consider to specify exactly which are the main statistics that you considered for each analysis.

P7, line 12. Please write here the acronym for the root mean square error.

P7, line 18. These results were not introduced in the methods. How did you generated this geomorphological maps? Please provide more details. In addition, since you mentioned the hillshaded DEM, I suggest again to revisit the manuscript and modifying the order of the analyses and corresponding results by describing firstly the DEM data.

P7, line 19. Which field observations? Do you mean the targets measured during the flight? Please specify these observations in the method and for what analyses they were used. At page 8 and 9 you mention again the field measurements by comparing these observation with the results of horizontal displacement and sediment budget. This is not clear.

P7, line 28. Please use the acronym M3C2.

P8, line 13. The fluxes are well constrained by stable areas. Please, consider to better explain this statement.

P8, line 21. Add over the same area of interest. Change meters with m.

P8, line 26. "the absolute displacement of the frontal lobe of the earthflow is not properly captured, as the frontal lobe advanced by ca. 55 meters". Where can I see this observation? Is it possible to add a scale of the displacement vectors in the figures in order to have a clear view of the magnitude of the movement?

P8, line 31. Please clarify what do you mean with "the best estimate volume". I suggest to report also the information about the elevation changes in meters.

P9, line 26. raw images. Do you mean raw uncompressed image format or simply the image dataset.

P9, line 30. Actually one of the main drawback of the UAV photogrammetry is the need of GCPs to georeference the point cloud and often used to reduce possible systematic error that can occur especially in presence of flat terrain. Why do you compare here UAV-SfM with TLS but you not mention any comparison with terrestrial images or possible combination of ground-based acquisition with UAV in case of problems during the flight for example.

Some acronyms are introduced in the text like SfM, M3C2, UAV, DoD, 3D. Please, use them in whole text.

Figure 1. Please consider to add either a slope map or a DEM with contours of the study area.

Figure 7. Please use the same number of significant decimal places in the legend. I suggest for Table 3 and Table 4 to use 2 significant decimal like in Table 2.

Table 1. Some information are missing, like GSD and UAV details.

Table 4. The caption of the figure. The COSI-Corr algorithm is applied on the hillshaded and not "from the 3D point clouds"

Table 5. Please add information about the elevation change (e.g. mean and standard deviation).

*Reference:*
*Stumpf, A., Malet, J. P., Allemand, P., Ulrich, P. (2014). Surface reconstruction and landslide displacement measurements with Pléiades satellite images. ISPRS Journal of Photogrammetry and Remote Sensing, 95, 1-12.*

---

## Referee Comment (RC1) · Anonymous Referee #1 · 8 Jul 2017

This manuscript reports on the performance of UAV derived topography processed using SfM photogrammetry for the purpose of monitoring active slope processes in the foothills of the Swiss Central Alps. The manuscript is clearly written, well structured, and effectively documents the work undertaken by the authors. In this work, the authors apply COSI-Corr, M3C2 (Lague et al., 2013) and the GCD ArcGIS plugin (Wheaton et al., 2010) to report on the horizontal and 3D displacements, and sediment budget of the landslide complex. In my opinion, the combination of these three analyses provide a really robust characterisation of the short-term (inter-annual) dynamics of the earthflow investigated. Overall, I believe the manuscript could be suitable for publication, but the authors need to consider the main contribution of this work given: (1) the large body

of UAV and SfM research already published in Physical Geography facing academic journals; and (2) the now frequent use of UAVs for hillslope monitoring by geotechnical consultants. Specifically I think the following questions need to be addressed before this work is formally accepted for publication: In what ways does the 'performance of UAV for monitoring ground surface displacements' need further investigation? How does this work build on from the work of Lucieer et al. (2014) [Progress in Physical Geography] who also used multi-temporal UAV imagery and SfM to report on surface change, and displacement (using COSI-Corr) associated with landsliding? Is the value of this work related to the fact that it is a SfM case study or should the scientific findings regarding hillslope failure be more prominent in the manuscript? In places, the work would benefit from citing a wider range of up-to-date UAV and SfM articles, especially those pertaining to the application of UAVs to hillslope failure. This year alone a large number of highly relevant manuscripts have been published and should be acknowledged and discussed in the manuscript. This will allow the contribution/novelty of this research, beyond representing another potential 'application of SfM' case study, to be better communicated to practitioners within this rapidly developing area of remote sensing.

Some specific comments:

P1. Lines 25-26: SfM for multitemporal analysis is not in its early stages. There is now a vast body of research that addresses this topic.

P7. Lines 3-5: I see you did not survey the entire earthflow in 2013 and 2015? Is this not problematic for your assessment of the hillslopes sediment budget?

P3. Lines 19-26: How did you classify the different morphogenetic units? Please provide more detail on the geomorphological mapping in this research with reference to the approach undertaken to classify this particular hillslope failure (e.g. with reference to key geomorphological mapping literature). This information should be provided in the methods section. You could also, for example, use digitised morphogenetic zones

**ESurfD**
to produce a more detailed breakdown of geomorphological change using the 'budget segregation' feature in the ArcGIS GCD plugin provided by Wheaton et al. (2010).

P5. Lines 18-27: Did you use a multi-rotor or fixed wing aerial platform? What was the approximate distance between the camera and the surface of interest during image acquisition? Please add this detail and ensure all details pertaining to the camera settings are provided in the main body or appendix in line with the recommendations of O'Connor et al. (2017) [Progress in Physical Geography].

P6. Lines 25-28: Please provide more information on the errors associated with each raster surface used for differencing (beyond what is presented in section 3.1). The propagated error values used to threshold the DoD need to be presented alongside your results and in Table 5. What is the uncertainty (in  $\pm$  m3) associated with the estimates of erosion, deposition and the net volume of difference? How did you arrive at the minimum, best and maximum estimates – are they linked to your detection limits? Were these based on difference values used to threshold the DoDs? Did you use spot height checkpoints to derive propagated error values? You need to more clearly communicate these aspects in the manuscript.

P9 Lines 25-26: You suggest that your study "confirms that the SfM algorithm in itself is robust and can be applied to convert raw image datasets into very-high resolution 3D point clouds." This is rather obvious and has been documented and addressed in great detail in a vast number of published manuscripts. I think you might need to reconsider what the main findings of your work actually are – perhaps the scientific findings are more interesting than the methodological ones?

P9 Line 30-onwards: Is it worth commenting on the application of ground-control here and any influence control measurement may have had on the resulting pattern of morphological change? For example, did you have any issues placing GCPs on problematic terrain and did this impact your GCP spacing (suggested 25m spacing on P5. Line 25)? Does the GCP distribution weaken confidence in any of your findings? As I am

**ESurfD**
certain you are aware, the application of GCPs is a time-intensive process that is important for reducing uncertainty in topography surveys. These themes (amongst other aspects of the SfM workflow) have recently been addressed by the work of James et al. (2017) via articles published in the journals ESPL and Geomorphology. On inaccessible and unstable terrain ground control cannot always be applied for practical/safety reasons (e.g. volcanic terrain). There has been some discussion about the potential for using direct georeferencing based UAV-SfM workflows in hazardous terrain (e.g. Carbonneau and Dietrich, 2017, ESPL). I think you would benefit from acknowledging these approaches/methodological papers when discussing the merits of the UAV-SfM approach for monitoring earthflows in this manuscript. In summary, the latest SfM findings need to be better integrated into this manuscript.

P10 Lines 1-4: The regulatory framework for RPAS/UAV operation is rapidly evolving in many countries. Are you able to briefly highlight any specific considerations (with reference to support materials) pertinent to your work in Switzerland? I am sure this information will be beneficial to geoscientists/geomorphologists planning future work in Switzerland.

Table 1: It would be great to see the GCPs plotted in a figure so the reader can assess GCP distribution and the impacts it may have had on the quality of the surface reconstruction for each survey.

Technical corrections:

P2. Line 4: 'is' change to 'are'?

P4. Line 12: "auttaumn" change to autumn?

P5. Line 23: Change to "better capture complex 3D structures"?

P10 Line 5: Title for the next section is duplicated in the main body of section 4.1.

**Interactive comment on Earth Surf. Dynam. Discuss., https://doi.org/10.5194/esurf-2017-38,**

**ESurfD**
2017.

**ESurfD**

---

## Author Comment (AC1) · 8 Sep 2017

**REPLY TO COMMENTS OF REFEREE 1 FOR MANUSCRIPT esurf-2017-38**

We would like to thank anonymous referee #1 for the constructive comments and suggestions, which will help us to improve the manuscript. Below, we respond to the suggestions of referee #1.

**Comments to the Author**

*This manuscript reports on the performance of UAV derived topography processed using SfM photogrammetry for monitoring active slope processes in the foothills of the Swiss Central Alps. The manuscript is clearly written, well structured, and effectively documents the work undertaken by the authors. In this work, the authors apply COSI-Corr, M3C2 (Lague et al., 2013) and the GCD ArcGIS plugin (Wheaton et al., 2010) to report on the horizontal and 3D displacements, and sediment budget of the landslide complex. In my opinion, the combination of these three analyses provide a really robust characterisation of the short-term (inter-annual) dynamics of the earthflow investigated. Overall, I believe the manuscript could be suitable for publication, but the authors need to consider the main contribution of this work given: (1) the large body of UAV and SfM research already published in Physical Geography facing academic journals; and (2) the now frequent use of UAVs for hillslope monitoring by geotechnical consultants. Specifically I think the following questions need to be addressed before this work is formally accepted for publication: In what ways does the 'performance of UAV for monitoring ground surface displacements' need further investigation? How does this work build on from the work of Lucieer et al. (2014) [Progress in Physical Geography] who also used multi-temporal UAV imagery and SfM to report on surface change, and displacement (using COSI-Corr) associated with landsliding? Is the value of this work related to the fact that it is a SfM case study or should the scientific findings regarding hillslope failure be more prominent in the manuscript? In places, the work would benefit from citing a wider range of up-to-date UAV and SfM articles, especially those pertaining to the application of UAVs to hillslope failure. This year alone a large number of highly relevant manuscripts have been published and should be acknowledged and discussed in the manuscript. This will allow the contribution/novelty of this research, beyond representing another potential 'application of SfM' case study, to be better communicated to practitioners within this rapidly developing area of remote sensing.*

Our main objective (page 3, line 30) is landslide monitoring. For this purpose, we first introduce the different methodologies to get topographic data and discuss quickly the trade-off between accuracy and spatial range. Then, we give an overview of the recent literature dealing with natural hazard monitoring, based on topographic reconstructions, and finally come to SfM.

We agree that we need to clarify the purpose of this paper, and will do so in the revised manuscript. The novelty of our research is the combination of horizontal, 3D displacements and sediment budgets derived from high-resolution 3D point clouds to get an in-depth understanding of landslide mechanisms. Indeed, this paper is not only about monitoring landslides using an UAV-SfM framework, as it analyses in depth the potential of using time series of very-high resolution topographic reconstructions. Consequently, we will reformulate the introduction to clarify this.

We agree that there is now a large amount of scientific papers about the use of UAV-SfM framework to reconstruct topography. But, as far as we know, this is still not the case for the monitoring of dynamic environments, and especially mass movements, and – to our knowledge - few published papers exist on dense time-series of UAV-Sfm reconstructions.

**Some specific comments**

- *P1. Lines 25-26: SfM for multitemporal analysis is not in its early stages. There is now a vast body of research that addresses this topic.*

"in its early stages" is indeed overstated. This will be rephrased. However, we do think that temporal analysis of earth surface topography using UAV-SfM derived topographic reconstructions is not yet mainstream. Although several papers have shown the potential of UAV-SfM for monitoring physical processes through time, e.g. riverbeds dynamic, glaciers, landslides, a profound assessment of the potential of dense time-series for 4D monitoring of geomorphic phenomena is – to our knowledge – an important contribution to the research field.

- *P7. Lines 3-5: I see you did not survey the entire earthflow in 2013 and 2015? Is this not problematic for your assessment of the hillslopes sediment budget?*

The areas are not the same every year, as explained on page 7, line 2. As a consequence, we divided the computation of statistics into two parts. First, on the intersection of the three datasets to allow comparison of absolute values over the entire period. And secondly on the intersection of each spatial interval, in order to get the most information of each pair of datasets. Therefore, it is not a problem, just a small limitation in the assessment of the hillslope dynamics.

- *P3. Lines 19-26: How did you classify the different morphogenetic units? Please provide more detail on the geomorphological mapping in this research with reference to the approach undertaken to classify this particular hillslope failure (e.g. with reference to key geomorphological mapping literature). This information should be provided in the methods section. You could also, for example, use digitised morphogenetic zones to produce a more detailed breakdown of geomorphological change using the 'budget segregation' feature in the ArcGIS GCD plugin provided by Wheaton et al. (2010).*

The geomorphological map has been produced in order to properly sketch the configuration of the study area before presenting the main results of the paper. The geomorphological setting of the area is difficult to perceive for the reader, only based on a simple shaded DEM or an orthophoto. The content of the digital geomorphological map is based on expert knowledge, and aims to visualize the main parts of the earthflow based on Varnes (1978).

We are aware of this budget segregation in the GCD plugin but in our opinion, the geomorphological map that results from the GCD plugin provide a lot of detailed information, and is not ideal to introduce the overall geomorphological setting of the study area.

- *P5. Lines 18-27: Did you use a multi-rotor or fixed wing aerial platform? What was the approximate distance between the camera and the surface of interest during image acquisition? Please add this detail and ensure all details pertaining to the camera settings are provided in the main body or appendix in line with the recommendations of O'Connor et al. (2017) [Progress in Physical Geography].*

We used a custom Y6 multirotor with embedded DJI controllers, and flew on average at an altitude of ca. 60 m above the ground. We will add the camera settings in the revised manuscript, as recommended in O'Connor et al. (2017).

- *P6. Lines 25-28: Please provide more information on the errors associated with each raster surface used for differencing (beyond what is presented in section 3.1). The propagated error values used to threshold the DoD need to be presented alongside your results and in Table 5. What is the uncertainty (in ±m3) associated with the estimates of erosion, deposition and the net volume of difference? How did you arrive at the minimum, best and maximum estimates – are they linked to your detection limits? Were these based on*

*difference values used to threshold the DoDs? Did you use spot height checkpoints to derive propagated error values? You need to more clearly communicate these aspects in the manuscript.*

In Table 5, minimum and maximum estimates correspond to the estimate minus/plus the uncertainty, while the best estimate is the estimated value (see also Figure 6). All values are computed based on the thresholded DoDs, for which we first applied a uniform error surface on each DEM, i.e. the associated error to each topographic reconstruction presented in Table 1. We will explain this in more details in the revised version of the manuscript.

- *P9 Lines 25-26: You suggest that your study "confirms that the SfM algorithm in itself is robust and can be applied to convert raw image datasets into very-high resolution 3D point clouds." This is rather obvious and has been documented and addressed in great detail in a vast number of published manuscripts. I think you might need to reconsider what the main findings of your work actually are – perhaps the scientific findings are more interesting than the methodological ones?*

We agree with the point that SfM is now considered as robust for accurate 3D reconstruction of natural environments. As written earlier, we will refocus the discussion more on the scientific findings about hillslope failure mechanism and on the additional but complimentary value of very-high resolution 4D data for capturing landslide dynamics.

- *P9 Line 30-onwards: Is it worth commenting on the application of ground-control here and any influence control measurement may have had on the resulting pattern of morphological change? For example, did you have any issues placing GCPs on problematic terrain and did this impact your GCP spacing (suggested 25m spacing on P5. Line 25)? Does the GCP distribution weaken confidence in any of your findings? As I am certain you are aware, the application of GCPs is a time-intensive process that is important for reducing uncertainty in topography surveys. These themes (amongst other aspects of the SfM workflow) have recently been addressed by the work of James et al. (2017) via articles published in the journals ESPL and Geomorphology. On inaccessible and unstable terrain ground control cannot always be applied for practical/safety reasons (e.g. volcanic terrain). There has been some discussion about the potential for using direct georeferencing based UAV-SfM workflows in hazardous terrain (e.g. Carbonneau and Dietrich, 2017, ESPL). I think you would benefit from acknowledging these approaches/methodological papers when discussing the merits of the UAV-SfM approach for monitoring earthflows in this manuscript. In summary, the latest SfM findings need to be better integrated into this manuscript.*

The pattern of GCP is very regular, i.e. one GCP every ca. 25 m in each direction on the active area of the earthflow. A figure depicting the GCP pattern will be provided in the revised version of the manuscript. Even if the terrain was problematic/dangerous at some places, we managed to overcome this issue to put nearly all the GCP that we wanted, for the sake of accurate final 3D reconstructions. In fact, the parameter that mainly affects the accuracy is the error associated to the GPS measurements due to poor signal. This led us to remove some GCPs with high associated error, as errors propagate to the final global accuracy of the 3D reconstructions after point cloud georeferencing (Clapuyt et al., 2016). We will add an explanation about this in the revised version of the manuscript.

Direct georeferencing is of course a potential solution as long as you are able to embed an RTK GPS in the UAV platform to geotag each picture. Otherwise, in our opinion, it is still worth to take the time to measure GCPs manually, in order to have accurate outputs. At the time of the surveys, we had not yet integrated such a device in our UAV platform, especially because of the lack of suitable low-cost and lightweight RTK devices on the market at that time. But we will acknowledge recent papers about this issue in the introduction

- *P10 Lines 1-4: The regulatory framework for RPAS/UAV operation is rapidly evolving in many countries. Are you able to briefly highlight any specific considerations (with reference to support materials) pertinent to your work in Switzerland? I am sure this information will be beneficial to geoscientists/geomorphologists planning future work in Switzerland.*

As you mention, this type of legislation changes rapidly, so we prefer not include the specific rules for Switzerland as it is possible that they will change in the future. At the moment: under 30 kg, drones can be flown without a permit as long as the pilot maintains eye contact with the device.

Swiss aircraft regulation:

https://www.admin.ch/opc/en/classified-compilation/19940351/index.html.

Forbidden areas must be avoided:

(https://map.geo.admin.ch/?topic=aviation&bgLayer=ch.swisstopo.pixelkarte-grau&layers=ch.bafu.bundesinventare-vogelreservate,ch.bafu.bundesinventare-jagdbanngebiete,ch.bazl.einschraenkungen-drohnen&lang=en&layers_opacity=0.75,0.75,0.6&catalogNodes=1379)

- *Table 1: It would be great to see the GCPs plotted in a figure so the reader can assess GCP distribution and the impacts it may have had on the quality of the surface reconstruction for each survey.*

We will add a figure, which will contain the pattern of GCPs, along with the flight pattern (as requested by Referee #2).
* * *
Following technical corrections below will be changed in the text and do not need a specific answer at this moment of the review process.

**Technical corrections**

- *P2. Line 4: 'is' change to 'are'?*
- *P4. Line 12: "auttaumn" change to autumn?*
- *P5. Line 23: Change to "better capture complex 3D structures"?*
- *P10 Line 5: Title for the next section is duplicated in the main body of section 4.1.*
* * *
**References**

Clapuyt, F., Vanacker, V. and Van Oost, K.: Reproducibility of UAV-based earth topography reconstructions based on Structure-from-Motion algorithms, Geomorphology, 260, 4–15, doi:10.1016/j.geomorph.2015.05.011, 2016.

Varnes, D. J.: Slope Movement Types and Processes, Transp. Res. Board Spec. Rep., (176), 11–33, doi:In Special report 176: Landslides: Analysis and Control, Transportation Research Board, Washington, D.C., 1978.

---

## Author Comment (AC2)

**REPLY TO COMMENTS OF REFEREE 2 FOR MANUSCRIPT esurf-2017-38**

We would like to thank anonymous referee #2 for the constructive comments and suggestions, which will guide for the revisions. Below, we respond to the suggestions of referee #2.

**General comments/suggestions**

*In the manuscript, Clapuyt et al. present the results of UAV photogrammetric surveys carried out on an active landslide in Switzerland. Their results have been used to quantify the horizontal and the 3D ground surface displacements and the sediment budget of the landslide. The Authors focused the manuscript on the interpretation of the landslides dynamics based on the high resolution dataset provided by UAV photogrammetry and they used M3C2, COSI-Corr and GCD (ArcGis Plugin) to obtain a comprehensive analysis of the annual dynamic of the landslide.*

*The manuscript is well written and the work is very interesting and potentially useful for future developments of UAV photogrammetry for landslide monitoring. However, there are some points of the paper that require improvements. In my opinion, the manuscript require a minor revision before being considering for publication in ESurf. I include below some suggestions or comments that could be of interest for the authors to be incorporated in the final version of the manuscript.*

*1) Accuracy has been assessed comparing SfM photogrammetry with the ground control points used to georeferenced the dense point cloud. However, as visible in Figure 2, both surveys, 2013 and 2015, were able to cover some area outside the earthflow. In addition you used the digital elevation model (DEM) and the elevation difference on the common area to estimate the sediment budget. Why do not consider also the elevation difference between multi-temporal DEM on the stable areas outside the earthflow as additional analysis to evaluate the accuracy of the SfM reconstruction? This could be useful to evaluate the spatial distribution of elevation changes between the three survey campaigns.*

This is a very good idea, and we will certainly take it along for future work in the area. Unfortunately, the 3D point clouds that we have do not allow us to use the "stable ridges" around the earth flow to monitor differences in surface displacements. In fact, the area that we have monitored is small and centred on the earthflow. (1) The ground control points were regularly scattered over the active area and its very-near surroundings. Even if a larger area has been captured, mainly due to oblique photos, there is a lack of ground control points outside the active area to have a proper 3D reconstruction, which is necessary for this kind of accuracy assessment. Also, (2) pastures directly surrounding the active part of the flow are also slightly moving downwards and can not be considered to be "truly stable". (3) There are no distinctive features on our airphotos, e.g. massive boulders, roads, houses around the earthflow that can be considered as immobile over the period of interest.

*2) In my opinion, the Authors not emphasize the advantages of very high resolution UAV data (both point cloud and DEM) for the landslide monitoring in comparison with the state of the art and previous investigations done by Schwab et al. (2007) and Savi et al. (2013) on the same study area. In addition, I suggest to better highlight how the results derived by the three different methodologies can be combined and which improvement their combination can provide on the interpretation of the landslide dynamic.*

We agree with your comment. In the next version of the manuscript, we will discuss more extensively on the additional, but complimentary, value of very-high resolution 3D topographic data, with respect to previous research from Schwab et al. (2007) and Savi et al. (2013). In

addition, as the referee #1 also suggested, we will develop the discussion more on the scientific findings regarding hillslope failure and highlight the new findings about landslide dynamic understanding based on the combination of techniques and results.

*3) Concerning the structure of the paper I have some observations, starting from the introduction where, in my opinion, some information are missing and I found it a bit confused.*

*Introduction. The Authors report a general description of the high resolution techniques available for the reconstruction of earth's landform. Then, a sentence about the accuracy is provided, following by a more detailed description about spatial resolution and spatial extension for each sensor and platform. In the second paragraph, the concept of high resolution is repeat again regarding the landslides monitoring and surface displacements. In my opinion these two paragraphs, should be rewritten focusing on the target object, i.e. landslide monitoring, by giving a clear description of the advantages and disadvantages of different survey technologies currently used for landslides monitoring and surface displacements analysis. In the introduction, two times you wrote about the aspects that affect the choice of the technologies. In specific, at line 10 you mention that "the choice of the acquisition framework result from the trade-off between the spatial resolution needed and the extent of the study area", then at line 20 you mention that "return period for acquisition and the surveying cost remain important criteria for the selection of the data acquisition platforms". These two aspects ( i.e. return period and cost)also affects the acquisition framework. I assume that the resolution and return period for acquisition necessary for landslide monitoring are strongly site-specific and depends by the magnitude and the assessment of associated hazard. However, I suggest to consider to write which are the main parameters (like resolution, data type, weather, accuracy, location accessibility, spatial and temporal resolution, coverage, cost) to consider when making a choice between different high resolution technologies, focusing on the landslide monitoring. Since you mentioned in the text different technologies, please consider that in the last decade both satellite (i.e. very high resolution satellite imaging) and aerial imaging system have benefited from great technology improvements reaching similar sub-meter resolution. Recently Stumpf et al. (2017) investigated the potential of Pléiades satellite images for landslide monitoring. I suggest to describe the real advantages of UAV data like the 3D point cloud, cm resolution, etc. Maybe, I suggest to refer here the comparison of your study with the previous investigations done by Schwab et al.(2007), and Savi et al.(2103).*

The introduction will be reshaped according to both referee comments, as they are similar at some point. We will focus more on the actual subject of our research, i.e. landslide monitoring and its associated scientific findings using UAV-SfM framework. We will emphasize the advantages and drawbacks of using specifically this methodology for landslide monitoring, along with the main parameters to take into account when choosing one particular methodology. We missed the recent paper of Stumpf et al (2017), but we will include it as it is very relevant in our review on landslide monitoring techniques.

**Specific comments/suggestions**

- *P4, line 8. I consider inappropriate to add a sentence about the effect of climate change on landslide hazard at the end of the introduction and after the description of your work. Maybe consider to start the introduction with this general topic that help to focus the object of the manuscript.*

We agree with this comment. In fact, this paragraph was at the beginning of the manuscript in a previous draft. As written earlier, we will re-structure the introduction of the manuscript.

- *P4, line 2. First you computed the 3-dimensional surface displacement, then the horizontal and the sediment budget. Please change the order. In addition I suggest to introduce the acronym 3D at the beginning (P3, line 14) and use it in the entire text. If you consider to extend the DoD on the stable area in order to analysis the accuracy of the photogrammetric DEM, I suggest to firstly describe the sediment budget based on the DoD, then the COSI- Corr analysis and at the end the M3C2. This also because for the COSI-Corr the hillshaded DEM is the data input (P6, line17). Moreover, the DoD provides information mainly about the*

*vertical change, COSI-Corr the horizontal displacements and M3C2 is a full 3D analysis. I believe this sequence of the analyses more appropriate. If you change the order, then you should verify that you change it throughout the whole paper.*

We agree that this sequence is more logical and appropriate, i.e. from 1-D to 3-D. We will adapt the manuscript in that sense.

- *P5, line18. Please provide more information about the platform, data acquisition and processing like the type of the UAV platform, flight height and flight path, GSD for each epoch, number of oriented images and very important the locations of the GCPs on the survey area. If I understood well, the GCPs were used after the camera orientation to georeference the point cloud. Why you didn't consider to include some of these observations in the bundle block adjustment during the camera orientation and the remaining as check points? Perhaps worth a comment. Please consider to include in this section also the description of the DEM generation and the accuracy used for the different analysis. You explained that for the horizontal displacement a spatial resolution higher than 0.20 generated incoherent results. However, for the DoD you used 0.04 m cell size as the best possible spatial resolution. How did you estimated this, based on the GSD? Why not used the same cell size, considering that the mean annual horizontal displacement range between 5.7 m to a max. of 8.9 m? Usually photogrammetric point cloud is characterized by high noise. Did you remove the noise before to generate 4 cm resolution DEM.*

Some technical information are indeed missing because we did not want to overload the manuscript. However, as it is relevant, we will update Table 1 with the number of oriented images and ground sampling distance. We will add a figure depicting flight paths over the study area and location of ground control (or integrate it in Figure 2, if it can stay legible). In the methodology section, we will add details about the UAV platform, i.e. custom Y6 multirotor with embedded DJI controllers, and about the mean flight altitude, i.e. 60 m.

The choice of using all the GCPs for georeferencing was driven by the need to have the most accurate 3D reconstructions, as they are inputs for temporal analyses where errors propagates. As we already showed in a previous study (Clapuyt et al., 2016) that our methodology was accurate, we did not needed to perform a new analysis in that sense.

Raw point clouds from SfM reconstruction were filtered out before performing any further analysis. The resolution of the subsequent DEMs, i.e. 0.04 m, is estimated based on the average point density of the point clouds, in order to exploit the high-resolution character of the data without altering it by using interpolation between points. Besides, in order to compute the horizontal displacement with the image correlation algorithm, the resolution of shaded relief surfaces has been chosen after performing a sensitivity analysis on the parameters of the algorithm, i.e. resolution of the input, the window size, the step between each sliding window and the search range. As the study area has a very complex topography, the sensitivity analysis showed that image correlation worked best with a lower resolution as false positive correlation between features was minimal at the 20 cm resolution.

- *P7, line 18. These results were not introduced in the methods. How did you generated this geomorphological maps? Please provide more details. In addition, since you mentioned the hillshaded DEM, I suggest again to revisit the manuscript and modifying the order of the analyses and corresponding results by describing firstly the DEM data.*

We will introduce the making of the geomorphological map in the methods. We prefer to keep the geomorphological map at the beginning of the results, as we think that it provides essential

information on the geomorphological setting of the area that helps to get a clear picture of the study area and facilitate the interpretation of the results.

- *P8, line 26. "the absolute displacement of the frontal lobe of the earthflow is not properly captured, as the frontal lobe advanced by ca. 55 meters". Where can I see this observation? Is it possible to add a scale of the displacement vectors in the figures in order to have a clear view of the magnitude of the movement?*

We will add a figure showing the advance of the frontal lobe, based on the DEMs, from 2013 and 2014.
* * *
All other comments and suggestions below will be changed in the text and do not need a specific answer at this moment of the review process.

- *P5, line 11. Consider to change: Data acquisition and data processing.*
- *P5, lines 12-13. Is it really necessary "a 3D point cloud" or this is a sentence related to your study.*
- *P5, line 14. The acronym SfM is already introduced. In addition I cannot find the connection of this sentence where you introduce the SfM algorithms with the next one about the planning of the survey. Please clarify this sentence and consider to move it at line 28 when you introduce the SfM algorithms.*
- *P6, line 4. Consider to change the order of the analyses.*
- *P6, line 29. In order to avoid repetition, better to report the corresponding software in the specific paragraph. Please consider to specify exactly which are the main statistics that you considered for each analysis.*
- *P7, line 12. Please write here the acronym for the root mean square error.*
- *P7, line 19. Which field observations? Do you mean the targets measured during the flight? Please specify these observations in the method and for what analyses they were used. At page 8 and 9 you mention again the field measurements by comparing these observation with the results of horizontal displacement and sediment budget. This is not clear.*
- *P7, line 28. Please use the acronym M3C2.*
- *P8, line 13. The fluxes are well constrained by stable areas. Please, consider to better explain this statement.*
- *P8, line 21. Add over the same area of interest. Change meters with m.*
- *P8, line 31. Please clarify what do you mean with "the best estimate volume". I suggest to report also the information about the elevation changes in meters.*
- *P9, line 26. raw images. Do you mean raw uncompressed image format or simply the image dataset.*
- *P9, line 30. Actually one of the main drawback of the UAV photogrammetry is the need of GCPs to georeference the point cloud and often used to reduce possible systematic error that can occur especially in presence of flat terrain. Why do you compare here UAV-SfM with TLS but you not mention any comparison with terrestrial images or possible combination of ground-based acquisition with UAV in case of problems during the flight for example.*
- *Some acronyms are introduced in the text like SfM, M3C2, UAV, DoD, 3D. Please, use them in whole text.*
- *Figure 1. Please consider to add either a slope map or a DEM with contours of the study area.*
- *Figure 7. Please use the same number of significant decimal places in the legend. I suggest for Table 3 and Table 4 to use 2 significant decimal like in Table 2.*
- *Table 1. Some information are missing, like GSD and UAV details.*
- *Table 4. The caption of the figure. The COSI-Corr algorithm is applied on the hillshaded and not "from the 3D point clouds"*
- *Table 5. Please add information about the elevation change (e.g. mean and standard deviation).*

*Reference:*

*Stumpf, A., Malet, J. P., Allemand, P., Ulrich, P. (2014). Surface reconstruction and landslide displacement measurements with Pléiades satellite images. ISPRS Journal of Photogrammetry and Remote Sensing, 95, 1-12.*

**Reference**

Clapuyt, F., Vanacker, V. and Van Oost, K.: Reproducibility of UAV-based earth topography reconstructions based on Structure-from-Motion algorithms, Geomorphology, 260, 4–15, doi:10.1016/j.geomorph.2015.05.011, 2016.

---

## Author Response (AR1)

Dear reviewers,

Dear associate editor,

Dear editor,

We would like to thank both anonymous referees and associate editor for the constructive comments and suggestions, which helped us to improve the manuscript. Below, we respond to the suggestions of both reviewers and refer to modifications in the new version of the manuscript. Major changes are highlighted in the manuscript version below the response to reviewers.

Thank to reviewer's comments about the proper findings of our research, we modified the title of the manuscript. We therefore propose the following:

"Unravelling earthflow dynamics from 3D time-series analysis of UAV-SfM derived topographic models"

Sincerely yours,

François Clapuyt

**REPLY TO COMMENTS OF REFEREE 1**

**Comments to the Author**

*This manuscript reports on the performance of UAV derived topography processed using SfM photogrammetry for monitoring active slope processes in the foothills of the Swiss Central Alps. The manuscript is clearly written, well structured, and effectively documents the work undertaken by the authors. In this work, the authors apply COSI-Corr, M3C2 (Lague et al., 2013) and the GCD ArcGIS plugin (Wheaton et al., 2010) to report on the horizontal and 3D displacements, and sediment budget of the landslide complex. In my opinion, the combination of these three analyses provide a really robust characterisation of the short-term (inter-annual) dynamics of the earthflow investigated. Overall, I believe the manuscript could be suitable for publication, but the authors need to consider the main contribution of this work given: (1) the large body of UAV and SfM research already published in Physical Geography facing academic journals; and (2) the now frequent use of UAVs for hillslope monitoring by geotechnical consultants. Specifically I think the following questions need to be addressed before this work is formally accepted for publication: In what ways does the 'performance of UAV for monitoring ground surface displacements' need further investigation? How does this work build on from the work of Lucieer et al. (2014) [Progress in Physical Geography] who also used multi-temporal UAV imagery and SfM to report on surface change, and displacement (using COSI-Corr) associated with landsliding? Is the value of this work related to the fact that it is a SfM case study or should the scientific findings regarding hillslope failure be more prominent in the manuscript? In places, the work would benefit from citing a wider range of up-to-date UAV and SfM articles, especially those pertaining to the application of UAVs to hillslope failure. This year alone a large number of highly relevant manuscripts have been published and should be acknowledged and discussed in the manuscript. This will allow the contribution/novelty of this research, beyond representing another potential 'application of SfM' case study, to be better communicated to practitioners within this rapidly developing area of remote sensing.*

The novelty of our research is the combination of horizontal, 3D displacements and sediment budgets derived from high-resolution 3D point clouds to get an in-depth understanding of landslide mechanisms. Indeed, this paper is not only about monitoring landslides using an UAV-SfM framework, as it analyses in depth the potential of using time series of very-high resolution topographic reconstructions.

We agree that there is now a large amount of scientific papers about the use of UAV-SfM framework to reconstruct topography. But, as far as we know, this is not yet the case for monitoring of dynamic environments, and especially mass movements. To our knowledge, few peer-reviewed papers exist which exploit dense time-series of UAV-SfM reconstructions.

Following your recommendations, we structured the entire introduction with a better focus on the scientific findings of our research. See section 1, pp. 3-5. We also refocussed the discussion section in that sense.

**Some specific comments**

- *P1. Lines 25-26: SfM for multitemporal analysis is not in its early stages. There is now a vast body of research that addresses this topic.*

"in its early stages" is indeed overstated. We restructured the introduction section, and provide a review of relevant work using the UAV-SfM framework for landslide monitoring. See p. 4 lines 16-32 in particular. However, we do think that temporal analysis of earth surface topography using UAV-SfM derived topographic reconstructions is not yet mainstream. Although several papers have shown the potential of UAV-SfM for monitoring physical processes through time, e.g. riverbeds dynamic, glaciers, landslides, a profound assessment of

the potential of dense time-series for 4D monitoring of geomorphic phenomena is – to our knowledge – an important contribution to the research field.

- *P7. Lines 3-5: I see you did not survey the entire earthflow in 2013 and 2015? Is this not problematic for your assessment of the hillslopes sediment budget?*

The areas are not the same every year, as explained on page 8, lines 16-19. As a consequence, we divided the computation of statistics into two parts. First, on the intersection of the three datasets to allow comparison of absolute values over the entire period. And secondly on the intersection of each spatial interval, in order to get a maximum of information of each dataset pair. Therefore, it is not a problem, just a small limitation in the assessment of the hillslope dynamics, which has been explained and taken into account.

- *P3. Lines 19-26: How did you classify the different morphogenetic units? Please provide more detail on the geomorphological mapping in this research with reference to the approach undertaken to classify this particular hillslope failure (e.g. with reference to key geomorphological mapping literature). This information should be provided in the methods section. You could also, for example, use digitised morphogenetic zones to produce a more detailed breakdown of geomorphological change using the 'budget segregation' feature in the ArcGIS GCD plugin provided by Wheaton et al. (2010).*

The geomorphological map has been produced in order to properly sketch the configuration of the study area before presenting the main results of the paper. The geomorphological setting of the area is difficult to perceive for the reader, only based on a simple shaded DEM or an orthophoto. The content of the digital geomorphological map is based on expert knowledge, and aims to visualize the main parts of the earthflow based on Varnes (1978).

We are aware of the "budget segregation" tool of the GCD plugin. However, the aim of this section is to sketch the overall setting of the study area.

- *P5. Lines 18-27: Did you use a multi-rotor or fixed wing aerial platform? What was the approximate distance between the camera and the surface of interest during image acquisition? Please add this detail and ensure all details pertaining to the camera settings are provided in the main body or appendix in line with the recommendations of O'Connor et al. (2017) [Progress in Physical Geography].*

We used a custom Y6 multirotor with embedded DJI controllers, and flew on average at an altitude of ca. 60 m above the ground. We added this information on p. 6 line 11 and lines 17-24. Also, we present UAV and camera settings in the Table 1 following the guidelines recommended by O'Connor et al. (2017)

- *P6. Lines 25-28: Please provide more information on the errors associated with each raster surface used for differencing (beyond what is presented in section 3.1). The propagated error values used to threshold the DoD need to be presented alongside your results and in Table 5. What is the uncertainty (in ±m3) associated with the estimates of erosion, deposition and the net volume of difference? How did you arrive at the minimum, best and maximum estimates – are they linked to your detection limits? Were these based on difference values used to threshold the DoDs? Did you use spot height checkpoints to derive propagated error values? You need to more clearly communicate these aspects in the manuscript.*

In the former Table 5 (now Table 4), minimum and maximum estimates correspond to the estimate minus/plus the uncertainty, while the best estimate is the estimated value (see also Figure 5). However, we reformulated the results of the sediment budget by giving estimated values with the associated error (in ±m³). The text of section 3.3 has been adapted in that sense. All values are computed based on the thresholded DoDs, for which we first applied a uniform

error surface on each DEM, i.e. the associated error to each topographic reconstruction presented in Table 2. See p. 7 lines 12-17 in the methodology section, and p. 9 lines 12-15 and lines 20-21 in the result section 3.3.

- *P9 Lines 25-26: You suggest that your study "confirms that the SfM algorithm in itself is robust and can be applied to convert raw image datasets into very-high resolution 3D point clouds." This is rather obvious and has been documented and addressed in great detail in a vast number of published manuscripts. I think you might need to reconsider what the main findings of your work actually are – perhaps the scientific findings are more interesting than the methodological ones?*

We agree with the point that SfM is now considered to be robust for accurate 3D reconstruction of natural environments. As written earlier, we refocussed the discussion more on the scientific findings about hillslope failure mechanism and on the additional but complimentary value of very-high resolution 4D data for capturing internal landslide dynamics. We therefore rephrased and adapted the discussion section 4.1 on p. 11 lines 5-24 and extended the discussion of scientific findings in section 4.2, p. 12 lines 8-22.

- *P9 Line 30-onwards: Is it worth commenting on the application of ground-control here and any influence control measurement may have had on the resulting pattern of morphological change? For example, did you have any issues placing GCPs on problematic terrain and did this impact your GCP spacing (suggested 25m spacing on P5. Line 25)? Does the GCP distribution weaken confidence in any of your findings? As I am certain you are aware, the application of GCPs is a time-intensive process that is important for reducing uncertainty in topography surveys. These themes (amongst other aspects of the SfM workflow) have recently been addressed by the work of James et al. (2017) via articles published in the journals ESPL and Geomorphology. On inaccessible and unstable terrain ground control cannot always be applied for practical/safety reasons (e.g. volcanic terrain). There has been some discussion about the potential for using direct georeferencing based UAV-SfM workflows in hazardous terrain (e.g. Carbonneau and Dietrich, 2017, ESPL). I think you would benefit from acknowledging these approaches/methodological papers when discussing the merits of the UAV-SfM approach for monitoring earthflows in this manuscript. In summary, the latest SfM findings need to be better integrated into this manuscript.*

The pattern of GCP is very regular, i.e. one GCP every ca. 25 m in each direction on the active area of the earthflow, which is now presented in Figure 2. Even if the terrain was problematic/dangerous at some places, we managed to overcome this issue to put nearly all the GCP that we wanted, for the sake of accurate final 3D reconstructions. In fact, the parameter that mainly affects the accuracy is the error associated to the GPS measurements due to poor signal. This led us to remove some GCPs with high associated error, as errors propagate to the final global accuracy of the 3D reconstructions after point cloud georeferencing (Clapuyt et al., 2016). We added this information in the methodology section on p. 6 lines 32-33 and p. 7 lines 1-2.

Direct georeferencing is of course a potential solution as long as you are able to embed an RTK GPS in the UAV platform to geotag each picture. Otherwise, in our opinion, it is still worth to take the time to measure GCPs manually, in order to have accurate outputs. At the time of the surveys, we had not yet integrated such a device in our UAV platform, especially because of the lack of suitable low-cost and lightweight RTK devices on the market at that time. We added a piece of discussion about recent developments in direct georeferencing research, along with references that you mention. See p. 11 lines 13-24 in the discussion section 4.1.

- *P10 Lines 1-4: The regulatory framework for RPAS/UAV operation is rapidly evolving in many countries. Are you able to briefly highlight any specific considerations (with reference to support materials) pertinent to your work in Switzerland? I am sure this information will be beneficial to geoscientists/geomorphologists planning future work in Switzerland.*

As you mention, this type of legislation changes rapidly, so we prefer not include the specific rules for Switzerland as it is possible that they will change in the near future. At the moment: under 30 kg, drones can be flown without a permit as long as the pilot maintains eye contact with the device, outside of forbidden areas, i.e. mainly airports and military areas. For your information, the Swiss aircraft regulation can be found here:https://www.admin.ch/opc/en/classified-compilation/19940351/index.html. We added a reference which reviews the current state of UAV regulations (Stöcker et al., 2017), page 11 line 29.

- *Table 1: It would be great to see the GCPs plotted in a figure so the reader can assess GCP distribution and the impacts it may have had on the quality of the surface reconstruction for each survey.*

We added Figure 2, which depicts flight paths of the UAV and GCP positioning over the earthflow area.

**Technical corrections**

- *P2. Line 4: 'is' change to 'are'?*

Change done, page 3 line 1.

- *P4. Line 12: "auttaumn" change to autumn?*

Change done, page 3 line 11.

- *P5. Line 23: Change to "better capture complex 3D structures"?*

Change done, page 6 line 17.

- *P10 Line 5: Title for the next section is duplicated in the main body of section 4.1.*

Duplicated title removed, page 11 line 31.

**References**

Clapuyt, F., Vanacker, V. and Van Oost, K.: Reproducibility of UAV-based earth topography reconstructions based on Structure-from-Motion algorithms, Geomorphology, 260, 4–15, doi:10.1016/j.geomorph.2015.05.011, 2016.

Stöcker, C., Bennett, R., Nex, F., Gerke, M. and Zevenbergen, J.: Review of the current state of UAV regulations, Remote Sens., 9(5), 33–35, doi:10.3390/rs9050459, 2017.

Varnes, D. J.: Slope Movement Types and Processes, Transp. Res. Board Spec. Rep., (176), 11–33, doi:In Special report 176: Landslides: Analysis and Control, Transportation Research Board, Washington, D.C., 1978.

**REPLY TO COMMENTS OF REFEREE 2**

**General comments/suggestions**

*In the manuscript, Clapuyt et al. present the results of UAV photogrammetric surveys carried out on an active landslide in Switzerland. Their results have been used to quantify the horizontal and the 3D ground surface displacements and the sediment budget of the landslide. The Authors focused the manuscript on the interpretation of the landslides dynamics based on the high resolution dataset provided by UAV photogrammetry and they used M3C2, COSI-Corr and GCD (ArcGis Plugin) to obtain a comprehensive analysis of the annual dynamic of the landslide.*

*The manuscript is well written and the work is very interesting and potentially useful for future developments of UAV photogrammetry for landslide monitoring. However, there are some points of the paper that require improvements. In my opinion, the manuscript require a minor revision before being considering for publication in ESurf. I include below some suggestions or comments that could be of interest for the authors to be incorporated in the final version of the manuscript.*

*1) Accuracy has been assessed comparing SfM photogrammetry with the ground control points used to georeferenced the dense point cloud. However, as visible in Figure 2, both surveys, 2013 and 2015, were able to cover some area outside the earthflow. In addition you used the digital elevation model (DEM) and the elevation difference on the common area to estimate the sediment budget. Why do not consider also the elevation difference between multi-temporal DEM on the stable areas outside the earthflow as additional analysis to evaluate the accuracy of the SfM reconstruction? This could be useful to evaluate the spatial distribution of elevation changes between the three survey campaigns.*

This is a very good idea, and we will certainly take it along for future work in the area. Unfortunately, the 3D point clouds that we have do not allow us to use the "stable ridges" around the earth flow to monitor differences in surface displacements. In fact, the area that we have monitored is small and centred on the earthflow. (1) The ground control points were regularly scattered over the active area and its very-near surroundings. Even if a larger area has been captured, mainly due to oblique photos, there is a lack of ground control points outside the active area to have a proper 3D reconstruction, which is necessary for this kind of accuracy assessment. Also, (2) pastures directly surrounding the active part of the flow are also slowly creeping downwards and cannot be considered to be "truly stable". (3) There are no distinctive features on our airphotos, e.g. massive boulders, roads, houses close to the earthflow that can be considered as immobile over the period of interest.

*2) In my opinion, the Authors not emphasize the advantages of very high resolution UAV data (both point cloud and DEM) for the landslide monitoring in comparison with the state of the art and previous investigations done by Schwab et al. (2007) and Savi et al. (2013) on the same study area. In addition, I suggest to better highlight how the results derived by the three different methodologies can be combined and which improvement their combination can provide on the interpretation of the landslide dynamic.*

We rephrased the objectives of the paper in the introduction section, p. 5 lines 1-9, to meet the real outcome of the research. We also added a discussion on the additional, but complimentary, value of very-high resolution 3D topographic data, with respect to previous research from Schwab et al. (2007) and Savi et al. (2013), on p. 12 lines 8-22.

*3) Concerning the structure of the paper I have some observations, starting from the introduction where, in my opinion, some information are missing and I found it a bit confused.*

*Introduction. The Authors report a general description of the high resolution techniques available for the reconstruction of earth's landform. Then, a sentence about the accuracy is provided, following by a more detailed description about spatial resolution and spatial extension for each sensor and platform. In the second paragraph, the concept of high resolution is repeat again regarding the landslides monitoring and surface displacements. In my opinion these two paragraphs, should be rewritten focusing on the target object, i.e. landslide monitoring, by giving a clear description of the advantages and disadvantages of different survey technologies currently used for landslides monitoring and surface displacements analysis. In the introduction, two times you wrote about the aspects that affect the choice of the technologies. In specific, at line 10 you mention that "the choice of the acquisition framework result from the trade-off between the spatial resolution needed and the extent of the study area", then at line 20 you mention that "return period for acquisition and the surveying cost remain important criteria for the selection of the data acquisition platforms". These two aspects ( i.e. return period and cost)also affects the acquisition framework. I assume that the resolution and return period for acquisition necessary for landslide monitoring are strongly site-specific and depends by the magnitude and the assessment of associated hazard. However, I suggest to consider to write which are the main parameters (like resolution, data type, weather, accuracy, location accessibility, spatial and temporal resolution, coverage, cost) to consider when making a choice between different high resolution technologies, focusing on the landslide monitoring. Since you mentioned in the text different technologies, please consider that in the last decade both satellite (i.e. very high resolution satellite imaging) and aerial imaging system have benefited from great technology improvements reaching similar sub-meter resolution. Recently Stumpf et al. (2017) investigated the potential of Pléiades satellite images for landslide monitoring. I suggest to describe the real advantages of UAV data like the 3D point cloud, cm resolution, etc. Maybe, I suggest to refer here the comparison of your study with the previous investigations done by Schwab et al.(2007), and Savi et al.(2103).*

We restructured the introduction according to the referees' comments, as they are similar at some point. We focussed more on the actual subject of our research, i.e. landslide monitoring and its associated scientific findings using UAV-SfM framework. We now review topography data acquisition techniques related to landslide monitoring, along with relevant research, see p3 lines 26-33 and p.4 lines 1-14. We added the parameters to consider when choosing one methodology, on p. 3 lines 21-24.

**Specific comments/suggestions**

- *P4, line 8. I consider inappropriate to add a sentence about the effect of climate change on landslide hazard at the end of the introduction and after the description of your work. Maybe consider to start the introduction with this general topic that help to focus the object of the manuscript.*

We moved this section about general context at the beginning of the introduction, as we agree with this comment.

- *P4, line 2. First you computed the 3-dimensional surface displacement, then the horizontal and the sediment budget. Please change the order. In addition I suggest to introduce the acronym 3D at the beginning (P3, line 14) and use it in the entire text. If you consider to extend the DoD on the stable area in order to analysis the accuracy of the photogrammetric DEM, I suggest to firstly describe the sediment budget based on the DoD, then the COSI- Corr analysis and at the end the M3C2. This also because for the COSI-Corr the hillshaded DEM is the data input (P6, line17). Moreover, the DoD provides information mainly about the vertical change, COSI-Corr the horizontal displacements and M3C2 is a full 3D analysis. I believe this sequence of the analyses more appropriate. If you change the order, then you should verify that you change it throughout the whole paper.*

We changed the order of results presentation, as it is indeed more logical and appropriate, i.e. from 1-D to 3-D. The order of associated tables and figures has been adapted in that sense too.

- *P5, line18. Please provide more information about the platform, data acquisition and processing like the type of the UAV platform, flight height and flight path, GSD for each epoch, number of oriented images and very important the locations of the GCPs on the survey area. If I understood well, the GCPs were used after the camera orientation to georeference the point cloud. Why you didn't consider to include some of these observations in the bundle block adjustment during the camera orientation and the remaining as check points? Perhaps worth a comment. Please consider to include in this section also the description of the DEM generation and the accuracy used for the different analysis. You explained that for the horizontal displacement a spatial resolution higher than 0.20 generated incoherent results. However, for the DoD you used 0.04 m cell size as the best possible spatial resolution. How did you estimated this, based on the GSD? Why not used the same cell size, considering that the mean annual horizontal displacement range between 5.7 m to a max. of 8.9 m? Usually photogrammetric point cloud is characterized by high noise. Did you remove the noise before to generate 4 cm resolution DEM.*

We followed the recommendations of O'Connor et al. (2017) about the required parameters to be included in publications using UAV-SfM methodology. All the camera and flight parameters are presented in Table 1, which now complements Table 2, focussed on point clouds characteristics. We also added the information about the UAV platform specification in the methodology section, p. 6 line 11.

The choice of using all the GCPs for georeferencing was driven by the need to have the most accurate 3D reconstructions, as they are inputs for temporal analyses where errors propagates. As we already showed in a previous study (Clapuyt et al., 2016) that our methodology was accurate, we did not needed to perform a new analysis in that sense. We now provide Figure 2 which depict GCP locations and flights paths, with the example of October 2015. In fact, we exactly repeated the methodology established in Clapuyt et al. (2016) for the 3 acquisition dates. The only parameter that is varying is the final number of GCPs taken into account for georeferencing (Table 2) because we discarded observations with a high associated error due to GPS signal weakness.

Regarding DSM generation and accuracy assessment: Raw point clouds from SfM reconstruction were filtered out before performing any further analysis. The resolution of the subsequent DEMs, i.e. 0.04 m, is estimated based on the average point density of the point clouds, in order to exploit the high-resolution character of the data without altering it by using interpolation between points. This is now explained in the methodology section, p. 6 lines 32-33 and p. 7 lines 1-5. The accuracy associated to each DSM is the one associated to point clouds. Besides, in order to compute the horizontal displacement with the image correlation algorithm, the resolution of shaded relief surfaces has been chosen after performing a sensitivity analysis on the parameters of the algorithm, i.e. resolution of the input, the window size, the step between each sliding window and the search range. As the study area has a very complex topography, the sensitivity analysis showed that image correlation worked best with a lower resolution as false positive correlation between features was minimal at the 20 cm resolution. We added information in the results section on p. 9 lines 31-32 and p.10 line1.

- *P7, line 18. These results were not introduced in the methods. How did you generated this geomorphological maps? Please provide more details. In addition, since you mentioned the hillshaded DEM, I suggest again to revisit the manuscript and modifying the order of the analyses and corresponding results by describing firstly the DEM data.*

We added information on the creation of the geomorphological map on p. 8 lines 29-30 and p. 9 lines 1-2 in the section 3.2. We prefer to keep the geomorphological map at the beginning of the results, as we think that it provides essential information on the geomorphological setting of the area that helps to get a clear picture of the study area and facilitate the interpretation of the results. However, we precise that DSMs, hillshades and slope maps are outputs from the UAV-SfM data acquisition.

- *P8, line 26. "the absolute displacement of the frontal lobe of the earthflow is not properly captured, as the frontal lobe advanced by ca. 55 meters". Where can I see this observation? Is it possible to add a scale of the displacement vectors in the figures in order to have a clear view of the magnitude of the movement?*

We added Figure 9, which is an illustration of the frontal lobe shift between June 2014 and October 2015. We computed the shift of 55 m by measuring it based on the digital surface models.
* * *
**Technical corrections**

- *P5, line 11. Consider to change: Data acquisition and data processing.*

Changed on p. 6 line 5.

- *P5, lines 12-13. Is it really necessary "a 3D point cloud" or this is a sentence related to your study.*

Rephrased on p. 6 line 6.

- *P5, line 14. The acronym SfM is already introduced. In addition I cannot find the connection of this sentence where you introduce the SfM algorithms with the next one about the planning of the survey. Please clarify this sentence and consider to move it at line 28 when you introduce the SfM algorithms.*

Sentence moved down to p. 6 line 25.

- *P6, line 4. Consider to change the order of the analyses.*

Analyses were re-ordered, from 1-D to 3D results. Tables and figures were ordered accordingly.

- *P6, line 29. In order to avoid repetition, better to report the corresponding software in the specific paragraph. Please consider to specify exactly which are the main statistics that you considered for each analysis.*

We specified statistics for each analysis in the methodology section, p. 7 lines 15-17 for sediment budget, p. 7 lines 27-28 for horizontal displacements and p. 8 lines 4-5 for 3D point cloud comparison.

- *P7, line 12. Please write here the acronym for the root mean square error.*

Modified p. 8 line 22.

- *P7, line 19. Which field observations? Do you mean the targets measured during the flight? Please specify these observations in the method and for what analyses they were used. At page 8 and 9 you mention again the field measurements by comparing these observation with the results of horizontal displacement and sediment budget. This is not clear.*

We added precision about field observations in section 3.2, on p. 8 lines 29-30 and p. 9 lines 1-2.

- *P7, line 28. Please use the acronym M3C2.*

Modified p. 10 lines 23.

- *P8, line 13. The fluxes are well constrained by stable areas. Please, consider to better explain this statement.*

We specified this on p. 10 lines 6-7.

- *P8, line 21. Add over the same area of interest. Change meters with m.*

Modified on p. 10 lines 15-16.

- *P8, line 31. Please clarify what do you mean with "the best estimate volume". I suggest to report also the information about the elevation changes in meters.*

We now report elevations changes along with volumetric changes in the results section 3.5. These data are also shown in Table 4.

- *P9, line 26. raw images. Do you mean raw uncompressed image format or simply the image dataset.*

Sentence deleted when restructuring the discussion section.

- *P9, line 30. Actually one of the main drawback of the UAV photogrammetry is the need of GCPs to georeference the point cloud and often used to reduce possible systematic error that can occur especially in presence of flat terrain. Why do you compare here UAV-SfM with TLS but you not mention any comparison with terrestrial images or possible combination of ground-based acquisition with UAV in case of problems during the flight for example.*

- *Some acronyms are introduced in the text like SfM, M3C2, UAV, DoD, 3D. Please, use them in whole text.*

We standardized the use of acronyms in the entire text, and made sure to write the entire acronym at first use.

- *Figure 1. Please consider to add either a slope map or a DEM with contours of the study area.*

We added Figure 2, containing a hillshaded DEM, with contour lines (2m), also depicting flight paths and GCP locations.

- *Figure 7. Please use the same number of significant decimal places in the legend. I suggest for Table 3 and Table 4 to use 2 significant decimal like in Table 2.*

We used the same number of decimals, i.e. 2 significant digits, for all measurements in meter.

- *Table 1. Some information are missing, like GSD and UAV details.*

We added this information in a new table along with camera and flight parameters, i.e. Table 1. We preferred not to modify Table 2, which is exclusively related to point cloud characteristics.

- *Table 4. The caption of the figure. The COSI-Corr algorithm is applied on the hillshaded and not "from the 3D point clouds"*

We added these in formation in a new table along with camera and flight parameters, i.e. Table 1. See reply to comment above

- *Table 5. Please add information about the elevation change (e.g. mean and standard deviation).*

We added mean elevation changes in meter, with associated error, for each time interval and area of interest in Table 4.

**Unravelling earthflow dynamics from 3D time-series analysis of UAV-SfM derived topographic models**

François Clapuyt[1], Veerle Vanacker[1], Fritz Schlunegger[2], Kristof Van Oost[1]

[1]Earth and Life Institute, Georges Lemaître Centre for Earth and Climate Research, Université Catholique de Louvain, Belgium.

[2]Institut für Geologie, Universität Bern, Switzerland

*Correspondence to:* François Clapuyt (francois.clapuyt@uclouvain.be)

**Abstract.** Accurately assessing geohazards and quantifying landslide risks in mountainous environments gain importance in the context of the on-going global warming. For an in-depth understanding of slope failure mechanisms, accurate monitoring of the mass movement topography at high spatial and temporal resolutions remains essential. The choice of the acquisition framework for high-resolution topographic reconstructions will mainly result from the trade-off between the spatial resolution needed and the extent of the study area. Recent advances in the development of UAV-based (Unmanned Aerial Vehicle) image acquisition combined with Structure-from-Motion (SfM) algorithm for 3-dimensional (3D) reconstruction makes the UAV-SfM framework a competitive alternative to other high-resolution topographic techniques.

In this study, we aim at getting an in-depth knowledge of the Schimbrig earthflow located in the foothills of the Central Swiss Alps, by monitoring ground surface displacements at very high spatial and temporal resolution using the efficiency of the UAV-SfM framework. We produced distinct topographic datasets for three acquisition dates between 2013 and 2015 in order to conduct a comprehensive 3D analysis of the landslide. Therefore, we computed (1) the sediment budget of the hillslope, and (2) the horizontal and (3) the 3-dimensional surface displacements, and. The multitemporal UAV-SfM based topographic reconstructions allowed us to quantify rates of sediment redistribution and surface movements. Our data show that the Schimbrig earthflow is very active with mean annual horizontal displacement ranging between 6 and 9 meters. Combination and careful interpretation of high-resolution topographic analyses reveal the internal mechanisms of the earthflow and its complex rotational structure. In addition to variation in horizontal surface movements through time, we interestingly showed that the configuration of nested rotational units changes through time. Although there are major changes in the internal structure of the earthflow in the 2013-2015 period, the sediment budget of the drainage basin is nearly in equilibrium. As a consequence, our data show that the time lag between sediment mobilization by landslides and enhanced sediment fluxes in the river network can be considerable.

**Copyright Statement.** The Authors agree with the Licence and Copyright Agreement of the Earth Surface Dynamics Journal.

[revised manuscript text omitted]

---

## Editor Decision (ED1)

[revised manuscript text omitted]

Accumulation zone

Erosional escarpments

2013 - 2014

2014 - 2015

120 meters

500 meters

**Legend**

Surface subsidence

Surface bulging

Horizontal displacement

Sliding surface

Profile 2013

Reference profile 2014

Profile 2015

**Figure 10: Schematic summary of the internal dynamics of the Schimbrig earthflow based on very-high resolution monitoring using the UAV-SfM framework.**

**Tables**

Table 1: Camera and imaging characteristics for all the surveys.

| Camera Model | Canon EOS 550D |
|---|---|
| Lens Model | 28mm f/2.8 IS USM |
| Image resolution | 5184 * 3456 pixels |
| Crop factor | 1.6 |
| Approximate sensor size | 22.3 * 14.9 mm |
| Pixel pitch | 4.3 µm |
| Mean shutter speed | 1/1250 |
| Mean ISO | 388 |
| Mean f-number | 3.49 |
| Flight velocity | 2 m/s (idealised) |
| Flight height | 60 m (idealised) |
| Ground sample distance | 9.2 mm |

Table 2: Characteristics and accuracy assessment of 3D point cloud reconstructions. Root mean square errors (RMSE) are the standard deviation of differences between the coordinates of the ground control points, i.e. georeferencing targets, measured by GPS and the coordinates of these points within the 3D point clouds after georeferencing. Horizontal RMSE is computed by taking the horizontal components of the point coordinates while the vertical RMSE is computed by taking only the z component of point
10   coordinates.

| | | October 2013 | June 2014 | October 2015 |
|---|---|---|---|---|
| **Point cloud characteristics** | **Nb of pictures** | 1143 | 4519 | 2501 |
| | **Nb of GCPs** | 49 | 108 | 99 |
| | **Area (ha)** | 5.86 | 17.69 | 15.59 |
| | **Point density (pts/m²)** | **1456** | **1270** | **1080** |
| **RMSE (m)** | **Horizontal** | 0.23 | 0.20 | 0.20 |
| | **Vertical** | 0.06 | 0.05 | 0.08 |
| | **Total error** | **0.24** | **0.20** | **0.22** |

**Table 3: Limit of detection values for temporal analyses (meters).**

| | | 2013 - 2014 | 2014 - 2015 |
|---|---|---|---|
| **Limit of Detection (m)** | **Horizontal** | 0.30 | 0.28 |
| | **Vertical** | 0.08 | 0.09 |
| | **Total** | **0.31** | **0.30** |

**Table 4: Volumetric (cubic meters) and depth (meters) sediment budgets.**

| Area of interest | Intersection | | | | Interval | |
|---|---|---|---|---|---|---|
| Time interval | 2013-2014 | | 2014-2015 | | 2014-2015 | |
| | Estimate | Error (±) | Estimate | Error (±) | Estimate | Error (±) |
| **Total Volume of Erosion (m³)** | -11,345 | 3,118 | -9,054 | 2,329 | -15,093 | 4,098 |
| **Total Volume of Deposition (m³)** | 6,012 | 1,919 | 8,293 | 2,546 | 17,005 | 4,533 |
| **Total Net Volume of Difference (m³)** | -5,333 | 3,661 | -762 | 3,450 | 1,912 | 6,111 |
| **Average net thickness difference (m)** | -0.22 | 0.15 | -0.03 | 0.14 | 0.05 | 0.15 |

**Table 5: Mean annual horizontal displacement (meters) computed using the COSI-Corr algorithm applied on shaded relief surfaces.**

| Area of interest | Intersection | | Interval |
|---|---|---|---|
| Time interval | 2013-2014 | 2014-2015 | 2014-2015 |
| **Min.** | 0.47 | 0.23 | 0.23 |
| **1st Qu.** | 1.81 | 0.89 | 0.89 |
| **Median** | 3.74 | 2.20 | 3.62 |
| **Mean** | 8.91 | 5.74 | 6.30 |
| **3rd Qu.** | 13.06 | 11.25 | 11.84 |
| **Max.** | 40.16 | 21.06 | 21.06 |

**Table 6: Descriptive statistics of the distances between 3D point clouds (meters).**

| Area of interest | Intersection | | Interval |
|---|---|---|---|
| Time interval | 2013-2014 | 2014-2015 | 2014-2015 |
| **Min.** | -20.72 | -27.56 | -28.83 |
| **1st Qu.** | -1.06 | -1.00 | -0.89 |
| **Median** | -0.60 | -0.49 | -0.43 |
| **Mean** | -1.04 | -0.69 | -0.38 |
| **3rd Qu.** | 0.47 | 0.66 | 0.79 |
| **Max.** | 20.36 | 26.96 | 26.96 |

---

## Author Response (AR2)

Dear associate editor,

Dear editor,

We would like to thank the associate editor for the constructive comments and suggestions, which helped us to improve the manuscript again. Below, we respond to these suggestions and refer to modifications in the new version of the manuscript. Major changes are highlighted in the manuscript version below the response to associate editor.

On the suggestion of the associate editor, we slightly simplified the title of the manuscript for a better readability. We therefore propose the following:

"Unravelling earth flow dynamics with 3D time series derived from UAV-SfM models".

Sincerely yours,

François Clapuyt

**REPLY TO COMMENTS OF ASSOCIATE EDITOR**

**Comments to the Author**

*The authors provide a profound case study investigating a landslide with different methods using UAV image data. Two provided reviews, which are strongly appreciated, confirm the solid work done by the authors. The study can become an important resource for researchers studying landslides. However, before the manuscript can be published minor revision would be required.*

*The authors should put some more emphasis on the novel insight that they could gain with their approach. Thus, in the introduction chapter I suggest another (short) paragraph displaying the recent findings in landslide processes observations. Therefore, in the discussion chapter the authors could pick up on those findings to better display their studies benefit in this regard (and in general compare their own results to other landslide research), i.e. focusing on the internal dynamics (e.g. that an increase of earthflow activity has been measured but no more output into the river was observed).*

*Furthermore, the authors give additional information regarding monitoring methods for landslides in the introduction chapter as suggested by the reviews. At the moment these additions seem to miss some linkage to their own study. Thus, I suggest to reconsider these sections and more clearly highlight e.g. what previous studies could not comply/were missing to what the authors aimed for and thus more distinctly justify the authors methodological approach.*

*There are a few more minor comments that should be addressed, which are mentioned in the attached document.*

*I encourage the authors to correspond to these minor issues before the article can be published. Thank you for your consideration of this special issue and for your submission.*

To remain concise and focussed in the writing of the introduction, we added a small paragraph mentioning studies about landslide structure and dynamic using geophysical and geotechnical methods. See p. 5 lines 6-8. We come briefly back at this point in the discussion. See p. 12 lines 7-10.

Regarding the link between our study and the review of topographic methods, we added for each methodology the drawbacks of its use. It was in fact missing for TLS, SAR and LIDAR technologies. We then concluded the whole paragraph reviewing topographic methods by making the link with our own study. We state that for frequent data acquisition in relatively inaccessible areas, the use of a more flexible and low-cost method is indicated. See p. 4 lines 16-18.

The other minor comments are answered point by point below.

**Minor comments**

- *P1 line 1: I would suggest to modify the title some more for fluent readability. Maybe "Unravelling earth flow dynamics with 3D time series derived from UAV photogrammetry"?*

Thank you for the suggestion. We modified the title as you proposed but kept the term "SfM" for "UAV-SfM framework" and replaced "photogrammetry" by "models". See page 1 line 1.

- *P3 line 4: Rephrase to "…bedrock have thick interbedded mudstones, which increase conditions for landslide occurrence."*

Rephrased p. 3 line 4.

- *P3 line 32: and resolution ?*

Of course, resolution is also a limiting factor for classic aerial photogrammetry datasets. Rephrased p. 3 line 32.

- *P4 lines 4-32: Please, link this description stronger to your own study if possible. For instance, if those methods already exist, also for longer measuring periods, why did you not use them? What do these studies still miss compared to your approach for your specific case study?*

We added drawbacks for TLS, SAR and LIDAR methods which were missing. We then summarize the review of topographic methods at the end of the paragraph by stating that for our specific case, we need the UAV-SfM algorithm. See p. 4 lines 16-18.

- *P4 line 25: Rephrase to "from temporal pairs of topographic representations".*

Rephrased p. 4 line 29.

- *P6 line 8: Replace "latter" by "used camera".*

Change done p. 6 line 21.

- *P6 line 19: Replace "as" by "because".*

Change done p. 6 line 27.

- *P6 line 31: Rephrase to: "Some measurements were removed before final georeferencing because they had a high associated error due to bad GPS signal reception.".*

Rephrased p. 7 lines 9-11.

- *P7 line 7: Please, very shortly state why you aligned the data if it was already referenced?*

The text is not clear and leads to misunderstanding. In fact, we aligned the DSMs after point cloud interpolation, and not directly point clouds. We clarified and rephrased it p. 7 lines 15-17.

- *P9 lines 1-2: Rephrase to: "Field observations consist of GPS measurements of the scarp, flowlines and boundaries of the earthflow. Identification of stable ridges, and visual analysis of geomorphic units are based on expert knowledge."*

Rephrased p. 9 lines 8-10.

- *P11 line 15: Not necessarily. Well distributed GCPs around the area of interest with some GPCs within (especially for height adjustment) are important.*

Rephrased p. 11 lines 25-26.

- *P11 line 17: I have a small comment regarding this statement. GCPs do influence the results if they are included in the adjustment. However, they have no big influence if they are solely used for the Helmert transformation, which you chose in your study. Maybe rephrase the sentence to avoid misunderstanding.*

Rephrased p. 11 lines 27-29.

- *P11 line 24: Replace "study case" by "case study".*

Change done p. 12 line 4.

- *P12 line 11: Replace "when it comes to get" by "regarding".*

Change done p. 12 line 27.

- *P12 line 13: Rephrase to "However, we are now able to compute slip rates over the entire extent of the earthflow at very high temporal resolution applying image correlation algorithms to very high-resolution aerial images.".*

Rephrased p. 12 lines 29-30.

- *P14 lines 6-8: Please rephrase this final sentence because it is difficult to understand due to grammatical issues.*

Rephrased p. 14 lines 20-22.

**Unravelling earth flow dynamics with 3D time series derived from UAV-SfM models**

François Clapuyt[1], Veerle Vanacker[1], Fritz Schlunegger[2], Kristof Van Oost[1]

[1]Earth and Life Institute, Georges Lemaître Centre for Earth and Climate Research, Université Catholique de Louvain, Belgium.

[2]Institut für Geologie, Universität Bern, Switzerland

*Correspondence to:* François Clapuyt (francois.clapuyt@uclouvain.be)

**Abstract.** Accurately assessing geohazards and quantifying landslide risks in mountainous environments gain importance in the context of the on-going global warming. For an in-depth understanding of slope failure mechanisms, accurate monitoring of the mass movement topography at high spatial and temporal resolutions remains essential. The choice of the acquisition framework for high-resolution topographic reconstructions will mainly result from the trade-off between the spatial resolution needed and the extent of the study area. Recent advances in the development of UAV-based (Unmanned Aerial Vehicle) image acquisition combined with Structure-from-Motion (SfM) algorithm for 3-dimensional (3D) reconstruction makes the UAV-SfM framework a competitive alternative to other high-resolution topographic techniques.

In this study, we aim at getting an in-depth knowledge of the Schimbrig earthflow located in the foothills of the Central Swiss Alps, by monitoring ground surface displacements at very high spatial and temporal resolution using the efficiency of the UAV-SfM framework. We produced distinct topographic datasets for three acquisition dates between 2013 and 2015 in order to conduct a comprehensive 3D analysis of the landslide. Therefore, we computed (1) the sediment budget of the hillslope, and (2) the horizontal and (3) the 3-dimensional surface displacements, and. The multitemporal UAV-SfM based topographic reconstructions allowed us to quantify rates of sediment redistribution and surface movements. Our data show that the Schimbrig earthflow is very active with mean annual horizontal displacement ranging between 6 and 9 meters. Combination and careful interpretation of high-resolution topographic analyses reveal the internal mechanisms of the earthflow and its complex rotational structure. In addition to variation in horizontal surface movements through time, we interestingly showed that the configuration of nested rotational units changes through time. Although there are major changes in the internal structure of the earthflow in the 2013-2015 period, the sediment budget of the drainage basin is nearly in equilibrium. As a consequence, our data show that the time lag between sediment mobilization by landslides and enhanced sediment fluxes in the river network can be considerable.

**Copyright Statement.** The Authors agree with the Licence and Copyright Agreement of the Earth Surface Dynamics Journal.

[revised manuscript text omitted]